# Beyond Sample-Level Forgetting: Improving Reliability in Multimodal Unlearning

**Jianzhou Wang** [1 2] **Yirui Wu** [1 2] **Lixin Yuan** [1 3] **Wenxiao Zhang** [1 2] **Jun Liu** [4]

## Abstract

Multimodal unlearning aims to eliminate specific data from pretrained multimodal models, which offers significant advantages in data privacy and model efficiency. Current methods struggle to achieve the desired properties of effectiveness, reliability and locality, due to the complex inter-dependency of unimodal and multimodal knowledge. By introducing a causal perspective, we propose multimodal unlearning with decoupled knowledge components. To promote fine-grained understanding of multimodal context, we introduce Multimodal Variational Inference (MVI) to infer modal-specific and -consistent factors with incomplete sample observation. With foundation of decoupled knowledge, we propose contrastive semantic editing to regulate multimodal unlearning towards refined forgetting. Experiments on privacy- and copyright-sensitive scenarios validate effectiveness of our method across multiple scenarios, ensuring the unlearned model maintains high reliability and locality.

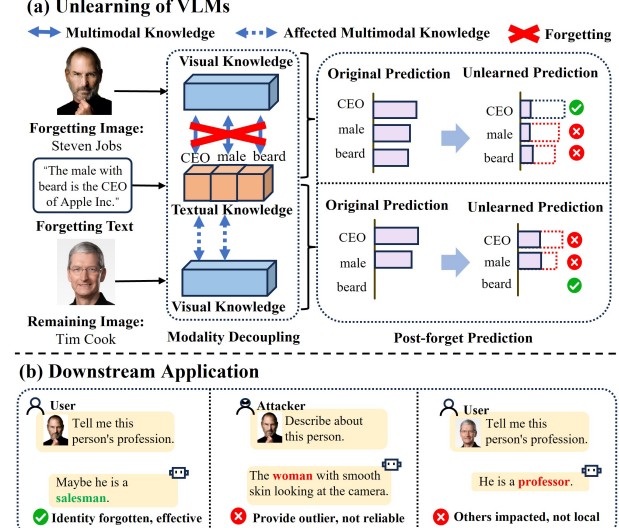

*Figure 1.* After forgetting, current multimodal unlearning methods predict with low reliability (i.e., wrong predictions of "male" and "beard" for Steven Jobs) and locality (i.e., low-confident predictions of "CEO" and "male" for Tim Cook), since their modality decoupling in sample-level semantics has an adverse impact on multimodal and unimodal knowledge.

## 1. Introduction

Multimodal models (Kim et al., 2021; Li et al., 2021; Liu et al., 2024a) have achieved modal-consistent understanding of the complex real-world, establishing across-modal relationship to perform varying down-stream tasks. Since training of multimodal models requires gathering vast user data, a major concern is arising that multimodal models may inadvertently infringe copyright and jeopardize individuals' privacy. Under regulations like GDPR (Mantelero, 2013),

service providers are obligated to eliminate specific data from both training datasets and model weights. Despite retraining from scratch on the remaining data meets the request, tremendous training consumption makes it unrealistic. There has been a further exploration, i.e., **multimodal unlearning**, which erases specific knowledge learned from the deleted samples (**Effectiveness**), while maintaining the capability of generating reasonable output given the forgotten samples as inputs (**Reliability**) without altering the output over other irrelevant inputs (**Locality**).

Leveraging unimodal knowledge (UK), i.e., modal-specified semantic representation encoded in parameters, and multimodal knowledge (MK), i.e., synergistic semantic representation to perceive correlations among modalities, Multi-delete (Cheng & Amiri, 2024) first decouples associations between unimodal samples with modality decoupling, and then retains UK and MK to preserve reliability. More recent works (Cai et al., 2025; Li et al., 2025) have employed localized gradient updates to balance the effectiveness of for-

---

[1]College of Computer Science and Software Engineering, Hohai University, Nanjing, China [2]Key Laboratory of Water Big Data Technology of Ministry of Water Resources, Hohai University, Nanjing, China [3]State Key Lab. for Novel Software Technology, Nanjing University, Nanjing, China [4]School of Computing and Communications, Lancaster University, Lancaster, UK. Correspondence to: Yirui Wu <wuyirui@hhu.edu.cn>.

*Proceedings of the $43^{rd}$ International Conference on Machine Learning*, Seoul, South Korea. PMLR 306, 2026. Copyright 2026 by the author(s).

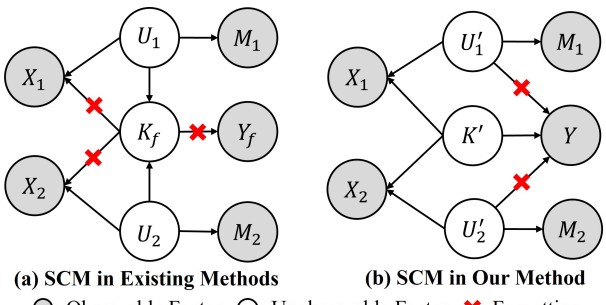

**(a) SCM in Existing Methods**  **(b) SCM in Our Method**

○: Observable Factor  ○: Unobservable Factor  ✗: Forgetting

*Figure 2.* Structural Causal Models of current methods (a) and our method (b).

getting with the preservation of remaining samples, thereby maintaining model locality. Despite achieving significant advances, their approaches of forgetting semantic correlations in sample level might impair **reliability** and **locality** of the original model. As shown in Figure 1 to forget Steven Jobs, current approaches fail in recognizing him as a bearded male with low reliability, due to the elimination of MK between his visual and textual embeddings. What's worse with the remaining sample Tim Cook, the post-forget model is unable to confirm him as *CEO* and *male* with low locality, since modality decoupling causes disorders on semantics, i.e., low confidence in the modified MK by altering the sharing semantics in UK about *CEO* and *male*. Both cases reveal current forgetting approaches cannot guarantee the effective performance with high reliability and locality.

In Figure 2(a), we look into causal relations of current methods (Cheng & Amiri, 2024; Cai et al., 2025; Li et al., 2025) using Structural Causal Model (SCM) (Pearl, 2010), including factors of input samples $X_i$, their corresponding modalities $M_i$, UK $U_i$, the forgetting MK $K_f$ and prediction $Y_f$. Multimodal unlearning expects to forget $(X_1, X_2)$ for risk-free prediction $Y_r = Y - Y_f$. By defining $i, j = 1, 2$ and $i \neq j$, $U_i \rightarrow X_i \& M_i$ represents the extraction of UK in sample level based on modality characteristics, $K_f \rightarrow X_i \& X_j$ represents the extraction of MK with the coupled samples based on different modalities, $K_f \rightarrow Y_f$ represents prediction based on MK, and $U_i \rightarrow K_f$ represents static dependency relation between UK and MK, since MK is built on perception of UK. Current methods decouple modalities by removing MK and maintaining UK to forget hidden relation between $X_1$ and $X_2$ in sample-level semantics. However, the existence of $U_i \rightarrow K_f$ would inadvertently cause the impairment of $U_i$ during the forgetting of $K_f$, leading to cases in Figure 1 by affecting UK of *CEO*, *male* and *beard*.

Beyond sample-level forgetting, we propose multimodal unlearning in modality decoupled semantics following SCM in Figure 2(b). As structured and semantically meaningful knowledge components, the learned representations can be edited rather than overwriting to reduce the risk of unreliability, and enforce cross-modal mappings across shared se-

mantic dimensions to minimize alignment bias. Specifically, by extracting modal-specific UK $U_i'$ and modal-consistent MK $K'$ as decoupled components, we build causal relation $U_i' \rightarrow Y$ to replace $U_i \rightarrow K_f \rightarrow Y_f$, **effectively** revealing causal effects from knowledge to prediction in a resolvable form, rather than the dependable one. Instead of forgetting $K_f$, we preserve $K'$ to **reliably** perform cross-modal comprehension, avoid introducing additional risk of privacy disclosure (Chundawat et al., 2023). More precisely, the post-forget model would produce Streisand Effect (Chen et al., 2021; Cooper et al., 2024) without $K'$, thus being vulnerable to membership inference attack. By independently extracting $U_i'$ from $X_i$ and $M_i$, we preserve **locality** of $U_i'$ without interference of relation $U_i \rightarrow K_f$. By editing fine-grained semantic components, we achieve a controllable direction to regulate multimodal forgetting with high effectiveness, reliability and locality.

In this paper, we propose a multimodal unlearning framework with knowledge decoupling and editing. We firstly design knowledge decoupling with Multimodal Variational Inference (MVI) to infer modal-specific semantic $U_i'$ and -consistent one $K'$ with incomplete observations. Then, we propose contrastive semantics editing to cut off $U_i' \rightarrow Y$ for effectiveness, maintain $K'$ and $U_i'$ to ensure reliability and locality respectively.

The contributions of our work are summarized as: **1)** Guided by causal analysis, we propose multimodal unlearning with decoupled multimodal and unimodal knowledge. **2)** We introduce MVI to infer modal-specific and -consistent factors with incomplete observation. **3)** We propose contrastive semantic editing to regulate multimodal unlearning towards refined forgetting. **4)** We evaluate effectiveness, reliability and locality in different scenarios to show our superiority.

## 2. Related Work

### 2.1. Machine Unlearning

Machine Unlearning aims to remove the influence of specific samples from a trained model without retraining, thereby balancing privacy and utility. Early weight-scrubbing methods (Golatkar et al., 2020; Guo et al., 2020; Foster et al., 2024) directly approximate how forgotten data affect model parameters, thus subtracting their influence via closed-form or iterative updates. Instead, optimization-based methods (Chourasia & Shah, 2023; Chundawat et al., 2023) fine-tune the model with mixture of the to-be-forgotten data and a subset of the remaining data, which erases unwanted information while preserving overall performance with carefully designed objectives. As model scales and data volumes grow especially in multimodal settings, more sophisticated techniques, such as gradient ascent reversal (Liu et al., 2022) and representation misdirection (Seo et al., 2024; Li et al., 2025),

have been developed, though avoiding collateral knowledge degradation remains challenging (Zhang et al., 2024).

## 2.2. Multimodal Pretrained Models

Multimodal Pretrained Models, particularly vision-language pretrained models (VLPs) (Kim et al., 2021; Radford et al., 2021; Li et al., 2021), learn to align and integrate representations across modalities (e.g., images and text) from large-scale, noisy web data. Their contrastive objectives enforce modal consistency by bringing paired inputs in latent space, while cross-modal interaction refines features for different downstream tasks like visual question answering (Antol et al., 2015) and image-text retrieval (Li et al., 2021). The vast and heterogeneous data used in VLP pretraining increases risks of encoding biases and privacy sensitivity, motivating to design unlearning and bias-mitigation strategies. While current model editing methods (Cheng et al., 2023; Pan et al., 2024; Shi et al., 2025) can correct explicitly identified errors, the scale and entanglement of multimodal knowledge make such assumptions difficult to satisfy under realistic, large-scale deletion requests.

## 3. Beyond Sample-level Unlearning

Inspired by the explanation on unlearning with decision boundary (Chen et al., 2023), we believe controllable multimodal unlearning can be accomplished by moving forgetting attributes out of the decision boundary. Following the causal guidance, the above assumption is hardly valid at first step, as the attribute-associated features are inherently obscured, or namely unseen in semantic space. Therefore, relocating boundary by few forgetting samples is unrealistic. Astute readers may intuitively realize it can be solved by decoupling: if every attribute is decoupled from knowledge, these unseen barriers would be removed. However, learning strictly decoupled attribute features is impossible without proper supervision (Locatello et al., 2019).

We propose another way around: **Multimodal Variational Inference** (MVI) to generate new samples, determining the joint distribution of grouped attributes as semantic components instead. More formally, we factorize $U_i'$ and $K'$ by $q(U_i', K'|X_i, Y, M_i) = q(U_i'|X_i, M_i)q(K'|Y)$ with observed $X_i$, $Y$ and $M_i$. It's worth noting $X_i$, $Y$ and $M_i$ provide sufficient constraints to rationally generate samples containing decoupled knowledge $U_i'$ and $K'$. The major difference between conventional knowledge decoupling and MVI-based generation is that, the former pursues *sample-agnostic* representation of attributes, while the latter encodes *sample-specific* semantic factors with the generated samples. Indeed, we alleviate the difficulties of strictly decoupling by making up the requested attributes in a simulation.

Within the decoupled knowledge, we further propose **Con-**

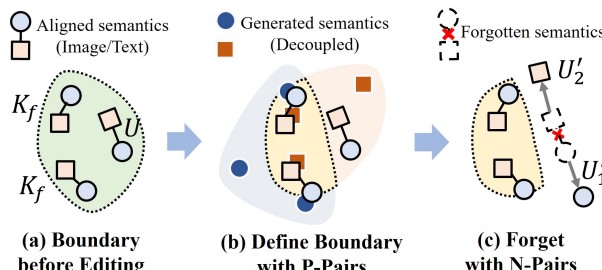

Figure 3. Contrastive Semantic Editing for controllable forgetting. (a) Semantic components from different modalities are aligned and located within sample-level decision boundary. (b) Positive pairs (P-Pairs) define new boundary around the overlap of decoupled semantics, moving $U$ out of the decision space. (c) Negative Pairs (N-Pairs) forget sample by taking apart different modalities.

**trastive Semantic Editing** to achieve forgetting in fine-grained level. Specifically, to build $U' \to Y$ for **effective** forgetting, we aim to establish a new decision boundary to replace sample-level one (Figure 3 (a)) for forgetting samples. Inspired by the high-quality feature representation capability of contrastive learning (Chen et al., 2020), we propose to achieve such boundary by pairing the forgetting and generated samples (Figure 3 (b)). In generated samples, decoupled semantic representations have differentiated distribution: the modal-consistent $K'$ overlaps, while modal-specific $U_i'$ naturally separate. Therefore, by pairing the forgetting sample with generated sample as Positive Pairs (P-Pairs) and penalizing their distance, we define a new boundary near the overlap. Such boundary mimics the decision behavior of retrained model, which moves $U$ out of the decision space and builds causal path $U_i' \to Y$. Afterwards, for complete forgetting via $U_i' \to Y$, we pair different modalities of forgetting sample as Negative Pairs (N-Pairs), penalizing similarity for separation (see Figure 3 (c)). Finally, we train P-Pairs and N-Pairs to predict high and low matching scores, respectively, thereby editing semantics according to $U_i' \& K' \to Y$.

To further refine $U_i' \& K' \to Y$, and preserve $K'$, $U_i'$ for **reliability**, **locality** respectively, we adjust the gradients during editing. Since all samples are driven by joint distribution $q(U_i', K')$, the disturbance of $U_i'$ and $K'$ prevent the prediction of $K' \to Y$ in positive pair and $U_i' \to Y$ in negative pair, respectively. We calculate similarity among samples with token-wise semantics comparing, which guides the reduction of noise brought by $U_i'$ or $K'$ in semantic space. More formally, we propose gradient-based optimization to shrink boundary, i.e., $G' = \alpha G(X_i, R_j) + (1 - \alpha)G(X_i, X_j)$, where $\alpha$ is the adjust vector to show the shrinking direction with similarity measuring, and $G()$ computes gradient for positive and negative pairs.

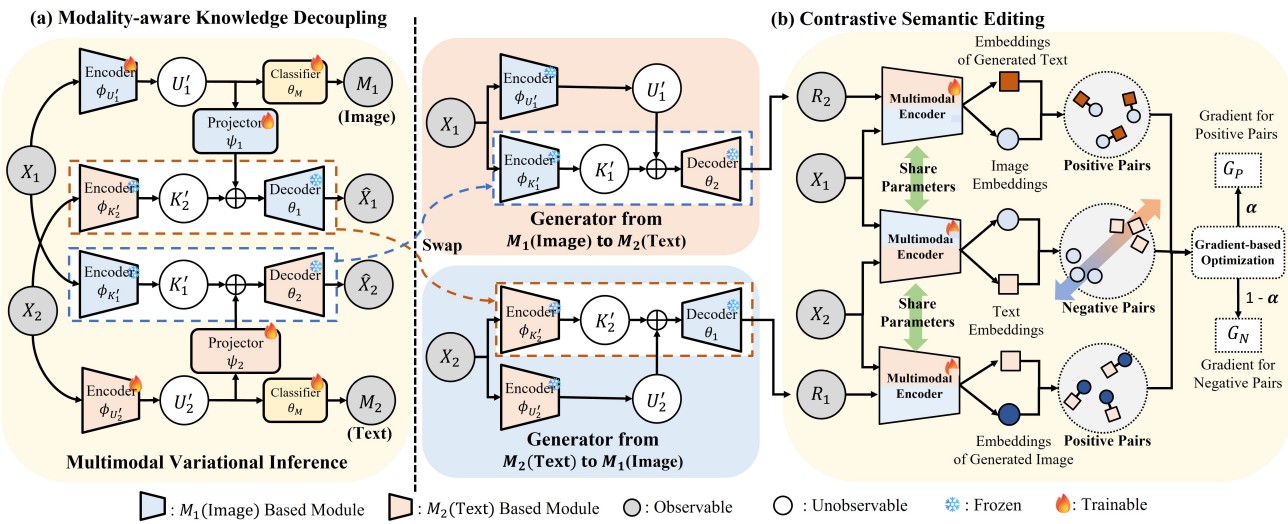

*Figure 4.* Structure overview: (a) Modality-aware knowledge decoupling, using Multimodal Variational Inference to generate latent representations of modal-specific semantics $U_i'$ and modal-consistent semantics $K_i'$; (b) Contrastive semantic editing, generating anchor samples $R_i$ based on decoupled $U_i'$, $K_i'$, and performing contrastive learning to controllably edit semantics of $X_i$ for precisely forgetting. Note that $K_i'$ refers to the specific multimodal knowledge, where $i = 1$ represents one from image to text, or vice versa.

## 4. Method

### 4.1. Structure Overview

In Figure 4, we illustrate our method with modality-aware knowledge decoupling and contrastive semantics editing. In decoupling, Multimodal Variational Inference(MVI) extracts $U_i'$ and $K'$ with inputting $X_i$. Specifically, we feed $X_i$ into frozen encoders $\phi_{U_i'}$ and learnable encoders $\phi_{K_i'}$ to extract $U_i'$ and $K_i'$, respectively. Then, $U_i'$ is sent to classifier to predict its belonging modality $M_i$, i.e., text or image, which optimizes $U_i'$ as supervised information. Afterwards, $U_i'$ is projected to the same space of $K_j'$ by projector, where they are concatenated as $[U_i', K_j'|i \neq j]$. Such vector is further used to reconstruct new sample $\hat{X}_i$ with decoder $\theta_i$. Finally, we minimize the difference between $\hat{X}_i$ and $X_i$, which is regarded as additional constraint to optimize $U_i'$ for its informativeness.

In editing, we adopt contrastive learning to define decision boundary with decoupled knowledge. By swapping $\phi_{K_i'}$ and $\theta_j$ of MVI, we transform such knowledge decoupling structure to cross-modal generators, which generates $M_1$(Image) to $M_2$(Text) or vice versa. Then, we generate samples $R_j$ with $[U_i', K_i']$ extracted from forgetting sample $X_i$ by decoupling. It's noted that both positive $(X_i, R_j)$ and negative pairs $(X_1, X_2)$ are encoded by a shared multimodal encoder. Afterwords, contrastive learning is used to pull the embeddings of positive pairs close, and push the embeddings of negative pair apart. After establishing decision boundary via contrastive learning, we propose gradient optimization to shrink the boundary with the adjusting vector $\boldsymbol{\alpha}$.

### 4.2. Modality-aware Knowledge Decoupling

We design Multimodal variational Inference(MVI) to decouple modality-conditioned knowledge, inferring approximate distribution of unobservable modal-specific and -consistent $U_i'$, $K'$ with constraints of observable $X_i$, $Y$ and $M_i$.

**General Structure.** Based on Markov conditions in SCM (Figure 2 (b)), the joint distribution contains all causal factors to be factorized as

$$P(X_i, Y, M_i, U_i', K') \\ = P(Y|K') \prod_{i=1}^{2} P(U_i', K')P(X_i|U_i', K')P(M_i|U_i'). \quad (1)$$

Guided by Equation (1), we design MVI as self-supervised encoder-decoder structure (Kingma & Welling, 2014), including encoders $\phi_{U_i'}$ and $\phi_{K_i'}$ to extract $U_i'$ and $K_j'$ by inferring $P(U_i', K') \approx q_{\phi_{U_i'}}(U_i'|X_i)q_{\phi_{K_j'}}(K_j'|X_i)$, decoder $\theta_i$ to reconstruct samples by learning $P(X_i|U_i', K') \approx p_{\theta_i}(X_i|\psi_i(U_i'), K_j')$, where $\psi_i$ is the projector that maps the knowledge of different modalities to the same dimension, and classifier $\theta_M$ to predict the modality by modeling $P(M_i|U_i') \approx p_{\theta_M}(M_i|U_i')$. Note that $P(Y|K')$ is implicitly formulated with $P(X_i|U_i', K')$, since prediction $Y \equiv X_i$ in self-supervised training. To sum up, MVI contains a set of network parameters to be determined $\{\phi_{U_i'}, \phi_{K_i'}, \theta_i, \psi_i, \theta_M|i = 1, 2\}$.

**Training Strategy.** To improve training efficiency, we employ the off-the-shelf pretrained generative models. Essentially, they excel at extracting modal-consistent semantics $K_j'$ by describing cross-modal transferring $p(X_i|K_j')$, where we can easily formulate their intermediate output

$K'_j$ as the condition to reconstruct cross-modal samples. Therefore, we utilize frozen parameters of the pretrained model to initialize $\phi_{K'_i}$ and decoders $\theta_i$. To train other parameters $\{\phi_{U'_i}, \psi_i, \theta_M\}$, we follow prior works (Blei et al., 2016; Hong et al., 2024) to maximize the evidence lower-bound(ELBO) from the observable factors $\{X_i, Y, M_i\}$:

$$ELBO(X_i, Y, M_i) = \sum_{i=1}^{2} \mathbb{E}_{U'_i}[\log p_{\theta_M}(M_i|U'_i)]$$
$$+ \sum_{\substack{i,j=1 \\ i \neq j}}^{2} \mathbb{E}_{U'_i, K'_j}[\log p_{\theta_i}(X_i|\psi_i(U'_i), K'_j)] \quad (2)$$
$$- \sum_{i=1}^{2} \mathbb{E}_{U'_i}[KL(p_{\phi_{U'_i}}(U'_i|X_i)\|P(U'_i))],$$

where $U'_i \sim q_{\phi_{U'_i}}(U'_i|X_i)$ and $K'_j \sim q_{\phi_{K'_j}}(K'_j|X_i)$ are the extracted latent factors by encoders $\phi_{U'_i}$ and $\phi_{K'_j}$, respectively. In Equation (2), the first term is the approximated log-likelihood of modality prediction based on the inferred modal-specific factors, and the second term equals reconstruction loss that jointly optimizes semantic extraction and sample generation. $KL(\cdot\|\cdot)$ denotes Kullback-Leibler (KL) divergence, which constrains distances of the inferred knowledge and their prior distribution, i.e., Multivariate Gaussian distribution $\mathcal{N}(\mathbf{0}, \mathbf{I})$ (Kingma & Welling, 2014). For details, see Appendix Section A.

**Analysis of knowledge decoupling.** $ELBO(X_i, Y, M_i)$ provides a computable lower-bound for marginal likelihood of observable factors. By maximizing $ELBO(X_i, Y, M_i)$, we optimize the inference capacity of the learnable parameters in MVI, thus decreasing uncertainty of distributions in Equation (1). By extracting $K'_j \sim q(K'_j|X_i)$ with frozen $\phi_{K'_j}$, learnable encoders $\phi_{U'_i}$ are forced to extract modality-specific semantics as $U'_i \sim q_{\phi_{U'_i}}(U'_i|X_i)$, where $U'_i$ and $K'_j$ are thus mutually complementary. To sum up, we use the posterior distribution $q(U'_i, K'|X_i) = q(U'_i|X_i)q(K'_j|X_i)$ to implicitly achieve practical and explainable knowledge decoupling, instead of strictly decoupling with sample-agnostic attributes.

### 4.3. Contrastive Semantic Editing

In Figure 5, we design contrastive semantic editing with steps of modality interaction and gradient-based optimization. For inputting, Positive Pair (P-Pair) includes the forgetting sample $X_i$ and $R_j$ generated by MVI based on $X_i$, meanwhile Negative Pair (N-Pair) refers to one sample with different modality representation, i.e., $X_i$ and $X_j$.

**Modality Interaction** includes 1) contrastive learning to pull P-Pair close and push N-Pair apart for cross-modal encoder optimization, and 2) cross-modal fusion to fuse paired embeddings as modal-agnostic representations.

In contrastive learning, we firstly encode $(X_i, R_j)$ and $(X_i, X_j)$ with multimodal encoder $\phi$, which computes image embedding $I_{X_i} = f_\phi(X_i) = [v_{cls}, v_l|l = 1, \ldots, L]$ and

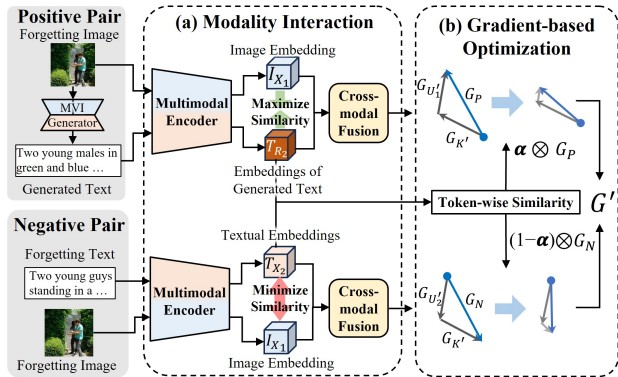

*Figure 5.* Structure design of contrastive semantic editing: (a) Modality interaction to perform contrastive leaning and cross-modal fusion, thus generating modal-agnostic representation for P-Pair and N-Pair; (b) Gradient optimization to relieve the disturbance of unimodal and multimodal knowledge on embedding encoding of P-Pair and N-Pair, respectively.

textual embedding $T_{X_i} = f_\phi(X_i) = [u_{cls}, u_k|k = 1, \ldots, K]$. It's noted that $v_{cls}$ and $u_{cls}$ denote the class token, and $L, K$ are numbers of tokens. Similarly, we define image and textual token of $R_j$ as $\hat{v}_l$ and $\hat{u}_k$, respectively. Afterwards, we optimize the encoding of $\phi$ to maximize and minimize distance of embeddings for P-Pairs $(X_1, R_2) \& (X_2, R_1)$ and N-Pairs $(X_1, X_2)$, respectively:

$$\mathcal{L}_{cl} = \frac{S(I_{X_1}, T_{X_2})}{\tau} - \frac{S(I_{X_1}, T_{R_2}) + S(I_{R_1}, T_{X_2})}{2\tau}, \quad (3)$$

where $\tau$ is the temperature parameter. We calculate similarity with $S(I_{X_i}, T_{R_j}) = f_I(v_l)^\top f_T(\hat{u}_k)$, where functions $f_I()$ and $f_T()$ perform linear transformation to ensure tokens as the same dimension. Essentially, such design encourages the encoder to cluster the shared semantics in P-Pairs and eliminate specific semantics in N-Pairs.

In cross-modal fusion, we adopt cross-modal weights to fuse the embeddings optimized by contrastive learning, obtaining representation for each pair in decision space. Essentially, we obtain an fused embedding with $E_{X_1, X_2} = f_F(f_{CA}(T_{X_2}, I_{X_1}, I_{X_1}))$, where functions $f_{CA}()$ and $f_F()$ refer to cross-attention scheme and feed forward network.

**Gradient-based Optimization** is adopted to optimize the fused embeddings, thus relieving the disturbance of unimodal and multimodal knowledge on embedding encoding of P-Pairs and N-Pairs, respectively. We firstly compute the gradient of fused embedding as $G = G_P + G_N = \frac{\partial(\mathcal{L}_{cl} + \mathcal{L}_{cf})}{\partial\Psi}$, where $G_P$ and $G_N$ are the gradients of P-Pairs and N-Pairs respectively, $\Psi$ is the parameters of the unlearning model. The cross-modal fusion loss $\mathcal{L}_{cf}$ is defined as

$$\mathcal{L}_{cf} = \log\left(\frac{2 \cdot p_y(I_{X_1}, T_{X_2})}{p_y(I_{X_1}, T_{R_2}) + p_y(I_{R_1}, T_{X_2})}\right), \quad (4)$$

where the comparison score between embedding is defined

*Table 1.* Results for privacy-sensitive scenario, where we report Membership Inference (MI, %), Recall (R, %), Accuracy (A, %), and Time (T, hours). $\Delta$ denotes the absolute difference to Retrain. Avg. G. represents the averaged absolute differences of $\Delta$. **Bold** indicates the best performance, and underline indicates the second best.

| Method | Flickr30k+MLLMU Retrieval | | | | | VQA+MLLMU Visual Multiple-choice Question | | | | |
| | MI ($\Delta\downarrow$) | R@$\mathcal{D}_f$ ($\Delta\downarrow$) | R@$\mathcal{D}_{test}$ ($\Delta\downarrow$) | Avg. G$\downarrow$ | T$\downarrow$ | MI ($\Delta\downarrow$) | A@$\mathcal{D}_f$ ($\Delta\downarrow$) | A@$\mathcal{D}_{test}$ ($\Delta\downarrow$) | Avg. G$\downarrow$ | T$\downarrow$ |
|---|---|---|---|---|---|---|---|---|---|---|
| | | | | | **5 Samples Unlearning** | | | | | |
| Retrain | 13.77 (0.00) | 14.38 (0.00) | 95.63 (0.00) | 0.00 | 69.50 | 12.65 (0.00) | 26.67 (0.00) | 63.14 (0.00) | 0.00 | 69.22 |
| Finetune | 14.59 (0.82) | 49.34 (34.96) | 92.06 (3.57) | 13.12 | 13.90 | 13.24 (0.59) | 66.67 (40.00) | 64.24 (1.10) | 13.90 | 12.53 |
| GA | 10.26 (3.51) | 54.26 (39.88) | 55.13 (40.50) | 27.96 | 0.01 | 12.02 (**0.63**) | 53.33 (26.66) | 57.92 (5.22) | 10.84 | 0.01 |
| BadT | 23.56 (9.79) | 34.48 (20.10) | 85.24 (10.39) | 13.43 | 0.02 | 23.10 (10.45) | 46.67 (20.00) | 52.77 (10.37) | 13.61 | 0.02 |
| Multidelete | 21.47 (7.70) | 33.25 (18.87) | 88.44 (7.19) | 11.25 | 0.02 | 21.87 (9.22) | 26.67 (**0.00**) | 56.49 (6.65) | 5.29 | 0.02 |
| SLUG | 17.25 (3.48) | 3.12 (11.26) | 89.75 (5.88) | 6.87 | 0.02 | 11.23 (1.42) | 33.33 (6.66) | 57.77 (5.37) | 9.01 | 0.02 |
| Adv | 11.24 (2.53) | 45.42 (31.04) | 84.79 (10.84) | 14.80 | 0.04 | 10.99 (1.66) | 33.33 (6.66) | 55.40 (7.74) | 5.35 | 0.03 |
| Ours | 14.02 (**0.25**) | 17.06 (2.68) | 90.09 (5.54) | **2.82** | 0.04 | 13.40 (0.75) | 26.67 (**0.00**) | 58.76 (**4.38**) | **1.71** | 0.03 |
| | | | | | **50 Samples Unlearning** | | | | | |
| Retrain | 13.52 (0.00) | 17.49 (0.00) | 95.63 (0.00) | 0.00 | 69.50 | 12.49 (0.00) | 28.64 (0.00) | 63.14 (0.00) | 0.00 | 69.22 |
| Finetune | 14.20 (0.68) | 50.14 (32.65) | 92.03 (3.60) | 12.31 | 13.91 | 14.10 (1.61) | 67.26 (38.62) | 64.22 (1.08) | 13.77 | 12.53 |
| GA | 10.32 (3.20) | 31.16 (13.67) | 31.49 (64.14) | 27.00 | 0.01 | 10.81 (1.68) | 26.32 (2.32) | 57.36 (5.78) | 3.26 | 0.01 |
| BadT | 24.24 (10.72) | 34.26 (16.77) | 75.27 (20.36) | 15.95 | 0.05 | 24.31 (11.82) | 38.52 (9.88) | 51.24 (11.90) | 11.20 | 0.06 |
| Multidelete | 23.46 (9.94) | 32.81 (15.32) | 72.39 (23.24) | 16.17 | 0.05 | 24.08 (11.59) | 26.44 (2.20) | 58.16 (4.98) | 6.26 | 0.06 |
| SLUG | 9.31 (4.21) | 30.73 (13.24) | 82.18 (13.45) | 10.30 | 0.06 | 15.62 (3.13) | 27.46 (1.18) | 58.29 (4.85) | 3.05 | 0.06 |
| Adv | 10.17 (3.35) | 43.13 (25.64) | 76.34 (19.29) | 16.09 | 0.22 | 10.75 (2.26) | 34.50 (5.86) | 56.32 (6.82) | 4.98 | 0.24 |
| Ours | 13.98 (**0.46**) | 16.82 (**0.67**) | 81.72 (13.91) | **5.01** | 0.22 | 13.73 (**1.24**) | 29.68 (1.04) | 59.22 (**3.92**) | **2.07** | 0.23 |
| | | | | | **500 Samples Unlearning** | | | | | |
| Retrain | 13.01 (0.00) | 15.21 (0.00) | 95.63 (0.00) | 0.00 | 69.50 | 10.40 (0.00) | 26.13 (0.00) | 63.14 (0.00) | 0.00 | 69.22 |
| Finetune | 18.92 (5.91) | 52.68 (37.47) | 91.81 (3.82) | 15.73 | 13.91 | 15.71 (5.31) | 60.01 (33.88) | 64.23 (1.09) | 13.43 | 12.51 |
| GA | 7.55 (5.46) | 18.26 (3.05) | 28.84 (66.79) | 25.10 | 0.09 | 6.94 (3.46) | 8.46 (17.67) | 50.40 (12.74) | 11.29 | 0.13 |
| BadT | 26.47 (13.46) | 36.18 (20.97) | 64.87 (30.76) | 21.73 | 0.49 | 26.34 (15.94) | 38.19 (12.06) | 55.64 (7.50) | 11.83 | 0.51 |
| Multidelete | 25.43 (12.42) | 32.45 (17.24) | 60.53 (35.10) | 21.59 | 0.65 | 25.06 (14.66) | 28.34 (2.21) | 53.46 (9.68) | 8.85 | 0.52 |
| SLUG | 14.37 (1.36) | 41.49 (26.28) | 64.31 (31.32) | 19.65 | 0.62 | 20.46 (10.06) | 26.91 (0.78) | 50.99 (12.15) | 22.99 | 0.51 |
| Adv | 9.86 (3.15) | 30.41 (15.20) | 63.76 (31.87) | 16.74 | 2.04 | 8.78 (**1.62**) | 20.24 (5.89) | 44.27 (18.87) | 8.79 | 2.08 |
| Ours | 12.03 (**0.98**) | 16.56 (**1.35**) | 71.49 (**24.14**) | **8.82** | 2.01 | 13.24 (2.84) | 26.89 (**0.76**) | 56.24 (**6.90**) | **3.50** | 2.06 |

as $p_y(I_{X_1}, T_{R_2}) = f_p(E_{X_1, R_2})$, $f_p()$ denotes the pre-trained classifier to predict the matching score. Recall the disturbance of $U_i'$ and $K'$ prevent the prediction of $K' \rightarrow Y$ in positive pair and $U_i' \rightarrow Y$ in negative pair respectively, we analyze the components of $G_P$ and $G_N$ with $G_P = G_{U_1'} + G_{K'}$ and $G_N = G_{U_2'} + G_{K'}$, which utilizes gradient view to evaluate the contribution of knowledge in establishing prediction.

Following the idea that uses token-wise semantics comparing to guide the noise reduction brought by $U_i'$ or $K'$, we compute token-wise similarity $\boldsymbol{\alpha} = \{\alpha_i | i = 1, ..., K\}$ based on $T_{X_2}$ and $T_{R_2}$:

$$\alpha_i = \max_{1 \le j \le K} s_{i,j}, \quad s_{i,j} = \frac{u_i \cdot \hat{u}_j}{\|u_i\| \|\hat{u}_j\|}, \ i, j \in [1, K], \quad (5)$$

where $u_i$ and $\hat{u}_i$ are tokens of $T_{X_2}$ and $T_{R_2}$ respectively. Essentially, we use $T_{X_2}$ and $T_{R_2}$ to define similarity, since both textual embeddings extracted from P-Pair and N-Pair are computed by the same encoder, thus offering a convinced way to compare.

Finally, we compute the optimized gradients with $G_{K'} = \boldsymbol{\alpha} \otimes G_P$ and $G_{U_i'} = (1 - \boldsymbol{\alpha}) \otimes G_N$, where $\otimes$ refers to element-wise multiplication. Such gradients can be used to optimize the prediction model.

## 5. Experiments

**Scenarios.** We consider 1) Privacy-sensitive, where training data includes a mixture of sensitive samples, e.g., personal biology or profiles and normal samples, e.g., natural images with descriptive captions; 2) Copyright-sensitive, where training data contains manually annotated image-text pairs, simulating that part of training data is required to be deleted due to copyright concerns.

**Tasks.** We adopt 1) Image-Text Retrieval: model retrieves the related texts with given images(TR) or vice verse(IR); 2) Visual Multiple-choice Question(V-MCQ): given an image and question about personal information, model predicts answers; 3) Visual Entailment(VE): regarding an image as premise and a sentence as hypothesis, model reasons whether hypothesis is entailment, neutral or contradiction.

**Datasets.** In privacy-sensitive scenario, we use Flickr30k (Young et al., 2014) and VQA (Antol et al., 2015) for retrieval and V-MCQ tasks respectively, and randomly select 5, 50 or 500 samples from MLLMU-Bench (Liu et al., 2024b) as $\mathcal{D}_f$ to simulate different unlearning scales. In copyright-sensitive scenario, we use Flickr30k and SNLI-VE (Xie et al., 2019) for retrieval and VE tasks respectively, and select 1K and 5K samples for different forgetting ratio following previous work (Cheng & Amiri, 2024).

*Table 2.* Results for copyright-sensitive scenario. **Bold** indicates the best performance, and underline indicates the second best.

| Method | Flickr30K Retrieval | | | | | SNLI-VE Visual Entailment | | | | |
|---|---|---|---|---|---|---|---|---|---|---|
| | MI ($\Delta \downarrow$) | R@$\mathcal{D}_f$ ($\Delta \downarrow$) | R@$\mathcal{D}_{test}$ ($\Delta \downarrow$) | Avg. G. $\downarrow$ | T $\downarrow$ | MI ($\Delta \downarrow$) | A@$\mathcal{D}_f$ ($\Delta \downarrow$) | A@$\mathcal{D}_{test}$ ($\Delta \downarrow$) | Avg. G. $\downarrow$ | T $\downarrow$ |
| **1K Samples Unlearning** | | | | | | | | | | |
| Retrain | 18.24 (0.00) | 94.57 (0.00) | 95.82 (0.00) | 0.00 | 65.23 | 17.41 (0.00) | 77.60 (0.00) | 80.27 (0.00) | 0.00 | 76.59 |
| Finetune | 20.11 (1.87) | 97.44 (2.87) | 95.81 (0.01) | 2.38 | 11.54 | 20.46 (3.05) | 88.83 (11.23) | 80.31 (0.04) | 4.77 | 14.75 |
| GA | 15.46 (2.78) | 73.10 (21.47) | 72.42 (23.40) | 15.88 | 0.31 | 15.75 (1.66) | 64.20 (13.4) | 58.49 (21.78) | 12.28 | 0.44 |
| BadT | 21.43 (3.19) | 52.83 (41.74) | 80.35 (15.47) | 20.13 | 1.70 | 23.41 (6.00) | 43.89 (33.71) | 75.04 (5.23) | 14.98 | 1.72 |
| Multidelete | 21.21 (2.97) | 54.10 (40.47) | 82.55 (13.27) | 18.90 | 1.73 | 24.73 (7.32) | 33.40 (44.20) | 78.75 (**1.52**) | 17.68 | 1.76 |
| SLUG | 20.85 (2.61) | 52.69 (41.88) | 71.98 (26.54) | 23.68 | 1.78 | 23.02 (5.61) | 32.95 (44.65) | 67.63 (12.64) | 20.97 | 1.76 |
| Adv | 14.56 (3.68) | 85.83 (8.74) | 94.20 (1.62) | 4.68 | 6.31 | 15.49 (1.92) | 34.27 (43.33) | 57.43 (22.84) | 22.70 | 6.41 |
| Ours | 17.76 (**0.48**) | 92.84 (**1.73**) | 94.47 (**1.35**) | **1.19** | 6.20 | 16.24 (**1.17**) | 68.24 (**9.36**) | 76.49 (3.78) | **4.77** | 6.35 |
| **5K Samples Unlearning** | | | | | | | | | | |
| Retrain | 18.26 (0.00) | 94.41 (0.00) | 95.33 (0.00) | 0.00 | 54.31 | 17.23 (0.00) | 77.60 (0.00) | 80.21 (0.00) | 0.00 | 62.49 |
| Finetune | 20.08 (1.82) | 97.51 (3.10) | 95.35 (0.02) | 1.65 | 11.52 | 19.90 (2.67) | 89.02 (11.42) | 80.30 (0.09) | 4.73 | 14.64 |
| GA | 13.24 (5.02) | 65.42 (28.99) | 61.05 (34.28) | 22.76 | 0.61 | 15.43 (1.80) | 62.74 (14.86) | 60.46 (19.75) | 12.14 | 0.86 |
| BadT | 21.42 (3.16) | 46.32 (48.09) | 78.25 (17.08) | 22.78 | 3.24 | 24.46 (7.23) | 42.33 (35.27) | 76.05 (4.16) | 15.55 | 3.34 |
| Multidelete | 20.75 (2.49) | 53.75 (40.66) | 82.55 (12.78) | 18.64 | 3.29 | 23.47 (6.24) | 34.20 (43.40) | 78.82 (**1.39**) | 17.01 | 3.48 |
| SLUG | 19.60 (1.34) | 50.91 (43.50) | 70.24 (25.09) | 23.31 | 3.25 | 22.44 (5.21) | 35.94 (41.66) | 68.76 (11.45) | 19.44 | 3.39 |
| Adv | 14.80 (3.46) | 86.59 (7.82) | 94.17 (1.16) | 4.15 | 12.36 | 16.26 (**0.97**) | 36.47 (41.13) | 56.21 (24.00) | 22.03 | 12.50 |
| Ours | 17.40 (**0.86**) | 92.33 (**2.08**) | 94.25 (**1.08**) | **1.34** | 11.92 | 16.20 (1.03) | 67.72 (**9.88**) | 76.61 (3.60) | **4.84** | 12.03 |

*Table 3.* Ablation study(%) on generation strategy in retrieval task with 50 forgetting samples. We report effectiveness $\mathcal{M}_{\text{eff}}$, reliability $\mathcal{M}_{\text{rel}}$ and locality $\mathcal{M}_{\text{loc}}$ as unlearning metrics, and Inception Scores (IS) Self-Bleu-4 (Self-B@4) as generation metrics.

| Generate Strategy | Privacy Sensitive | | | Copyright Sensitive | | | Quality Metrics | |
|---|---|---|---|---|---|---|---|---|
| | $\mathcal{M}_{eff}(\uparrow)$ | $\mathcal{M}_{rel}(\uparrow)$ | $\mathcal{M}_{loc}(\uparrow)$ | $\mathcal{M}_{eff}(\uparrow)$ | $\mathcal{M}_{rel}(\uparrow)$ | $\mathcal{M}_{loc}(\uparrow)$ | IS ($\uparrow$) | Self-B@4 ($\downarrow$) |
| Random selecting | 98.72 | 85.35 | **89.7** | 98.39 | 95.26 | **98.77** | - | - |
| Generate $R_j$ with only $U_i'$ | 99.49 | 76.60 | 75.28 | 99.47 | 93.60 | 86.34 | 0.22 | 78.07 |
| Generate $R_j$ with only $K_i'$ | 99.53 | 72.75 | 64.22 | 99.43 | 97.82 | 96.42 | 3.09 | 63.93 |
| Generate $R_j$ with $[U_j', K_i']$ | 98.24 | 78.46 | 62.57 | 97.69 | 96.76 | 89.6 | **3.11** | 63.72 |
| Generate $R_j$ with $[U_i', K_i']$ (**Ours**) | **99.54** | **99.33** | 86.09 | **99.52** | **98.27** | 98.65 | 3.08 | **60.67** |

**Model structures.** We employ ALBEF (Li et al., 2021) as primary foundational VLP for multimodal unlearning. To demonstrate the generality of our method, we further evaluate it on BLIP (Li et al., 2022) as an additional backbone, with detailed results reported in the Appendix B. We utilize Stable Diffusion (SD) (Rombach et al., 2022) as both text encoder and image decoder, and BLIP-2 (Li et al., 2023) as both image encoder and text decoder in MVI generator.

**Comparisons.** We compare with 1) Retrain (Huang et al., 2024) is the gold model in machine unlearning; 2) Finetune continuously finetunes the pre-trained model $\Psi$ on $D_r$ to forget $\mathcal{D}_f$ with parameter updating; 3) GA (Thudi et al., 2022) finetunes $\Psi$ on $\mathcal{D}_f$ with opposite gradient updating; 4) BadT (Chundawat et al., 2023) that finetunes the model with dual knowledge distillation; 5) Adv (Cha et al., 2024) forces the model to give incorrect predictions with adversarial attack; 6) MultiDelete (Cheng & Amiri, 2024) and 7) SLUG (Cai et al., 2025), two recent multimodal unlearning methods. For fair comparisons, we follow (Cheng & Amiri, 2024) to use the same access rate to $\mathcal{D}_r$ and forgetting rate.

**Metrics.** We use Membership Inference(MI) vulnerability (Shokri et al., 2017) for evaluation. We report the average of recall@$\{1, 5, 10\}$ on TR and IR for retrieval tasks, and accu-

racy for other tasks. We also report the absolute difference ($\Delta$) between methods and Retrain to measure effectiveness $\mathcal{M}_{\text{eff}} = 1 - \Delta MI$, reliability $\mathcal{M}_{\text{rel}} = 1 - \Delta R@\mathcal{D}_f$ and locality $\mathcal{M}_{\text{loc}} = 1 - \Delta R@\mathcal{D}_{test}$. Training time (T) is reported to measure efficiency of unlearning. To evaluate the quality of generated samples, we report Inception Scores(IS) (Goodfellow et al., 2020b) for generated images and Self-Bleu-4(Self-B@4) (Montahaei et al., 2019) for texts.

## 5.1. Main Results

**Comparison in privacy-sensitive scenario.** As shown in Table 1, our method outperforms existing approaches in most cases and achieves the smallest gap to Retrain. Finetune attains strong locality at high computational cost but shows poor reliability, indicated by a large gap in $R@\mathcal{D}_f$. Most baselines also suffer from insufficient forgetting, e.g., SLUG exhibits an average $16.93\%$ higher $R@\mathcal{D}_f$ than Retrain. In contrast, our method simultaneously achieves effectiveness, reliability, and locality with modest training overhead. Specifically, under 50-sample unlearning, it reduces the MI Gap by $2.74\%$, maintains reliability with only a $0.67\%$ gap in $R@\mathcal{D}_f$, and achieves the best locality with up to $5.38\%$ reduction in $\Delta R@\mathcal{D}_{test}$, indicating more precise semantic control during forgetting.

*Table 4.* Ablation study(%) for contrastive semantic editing. P-pairs, N-pairs and G-Opt denotes positive pairs, negative pairs, gradient-based optimization, respectively.

| P-Pairs | N-Pairs | G-Opt | $\mathcal{M}_{\text{eff}}(\uparrow)$ | $\mathcal{M}_{\text{rel}}(\uparrow)$ | $\mathcal{M}_{\text{loc}}(\uparrow)$ |
|:---:|:---:|:---:|:---:|:---:|:---:|
| ✓ | | | 95.82 | 96.3 | 85.43 |
| | ✓ | | 86.16 | 83.79 | 80.42 |
| ✓ | ✓ | | 88.12 | 90.53 | 66.38 |
| ✓ | ✓ | ✓ | **99.54** | **99.33** | **86.09** |

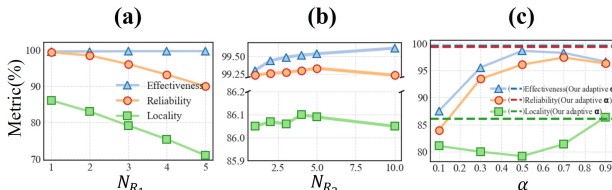

*Figure 6.* Ablation with different (a) generated images, (b) generated texts, (c) token-wise similarity $\alpha$.

**Comparison in copyright-sensitive scenario.** Due to naturally captured and well annotated, images in copyright-sensitive scenario generally share semantics, thus imposing challenges to maintain reliability. As shown in Table 2, advanced methods(e.g., SLUG) fail with low performance on $\mathcal{D}_f$, since they are designed with sample-level forgetting. In contrast, finetune overfits to $\mathcal{D}_f$ with high recall and accuracy, which proves multimodal knowledge learned from forgetting samples is insufficiently forgotten to decrease reliability. Although Adv achieves good performance in retrieval, it fails in VE, since its sample-level representation prevents proper transformation of relation among different modalities. Instead, our method exceeds other methods in all cases with nearly same performance with Retrain.

## 5.2. Ablation Studies

To investigate the effectiveness of different designs in our method, we perform ablation studies on Retrieval task in privacy-sensitive scenario 50 samples unlearning.

**Effect of Knowledge Decoupling and Generation Quality.** In Table 3, we study the effect of different combinations of decoupled knowledge used for generation. By selectively removing or swapping encoders of MVI, we can generate $R_i$ with different knowledge components. Randomly selecting $X_j$ as $R_j$ introduces additional $U'_j$ and improves locality, but reduces effectiveness and reliability due to insufficiently structured interactions between $U'_i$ and $K'$. In rows 2 and 3, the absence of either $U'_i$ or $K'$ decreases both reliability and locality, indicating that incomplete knowledge combinations fail to support semantic completeness. By canceling the swap operation as row 4, the generation is equal to $\hat{X}_i$ with worse locality, since $R_j$ is not generated with the proper $U'_i$, thus unable to define boundary in positive pairs. Our proposed strategy achieves the best performance to show its effectiveness with high generation quality. Moreover, privacy-sensitive data can be greatly affected by improper strategies, revealing that privacy-related semantics is embedded in $U'_i$. See Appendix for more qualitative examples and discussions about generation quality.

**Contrastive Semantic Editing.** As Table 4 shows, removing N-Pairs decreases effectiveness and reliability, since incomplete forgetting cause confusion among samples. When

P-Pairs are removed, $\mathcal{M}_{\text{eff}}$ and $\mathcal{M}_{\text{rel}}$ decrease sharply to 86.16% and 83.79%, due to failure in defining boundary without $R_i$. By removing gradient optimization, boundary would collapse with disorders in semantic space. Ours achieves best performance at the bottom row.

**Virtual Sample Numbers.** To investigate the sensitivity of the key hyperparameters controlling our framework, we conduct detailed ablation studies in Figure 6. In In Figure 6 (a), reliability and locality decreases when number of the generated images $N_{R_1}$ raises. With more generated images containing unobserved semantics, they may have few relevance with each other, where co-existence of multiple anchors results in conflicts of defining refined boundary. In Figure 6 (b), the number of generated text $N_{R_2}$ offers little influence on performance, since text can be well generated by informative images.

**Effect of Token-wise similarity $\alpha$.** In Figure 6 (c), we compare several values of $\alpha$. As observed, a fixed $\alpha$ struggles to keep balance among effectiveness, reliability, and locality, thus yielding suboptimal robustness. In contrast, our method with adaptive $\alpha$ ensures optimal performance in all metrics.

## 5.3. Qualitative Analysis

**Cross-modal matching score.** In Figure 7 (a), we visualize the confusion matrix before and after unlearning, where the diagonal is the matching score predicted with different forgetting samples. Observing high score for forgetting samples before unlearning, the post-forget model confuses the forgetting text with images with effective unlearning.

**Visualizations of cross-attention.** In Figure 7 (b), we use Grad-CAM (Li et al., 2021) to visualize the last layer of cross-attention in modality interaction. Before unlearning, the model focuses on facial features, such as eye spacing and lineament, regarding privacy samples. Such attention is distributed after forgetting, since our method effectively forget knowledge about privacy. Another example from $\mathcal{D}_r$ implies the post-forget model keeps the similar attention on the remembering objects with high reliability and locality.

**T-SNE visualization for semantic space.** In Figure 8, we visualize the semantic space using T-SNE. Before un-

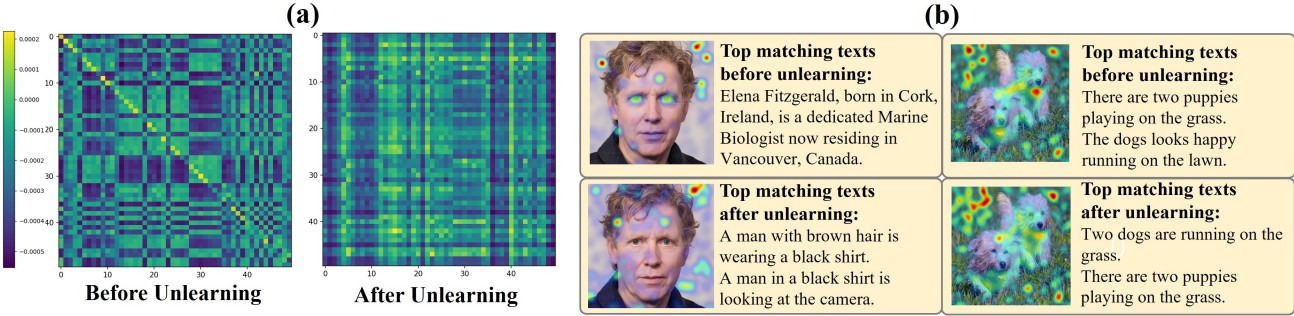

*Figure 7.* Qualitative results before and after unlearning. (a)Visualization of matching score matrix in privacy sensitive scenario. (b)Grad-CAM visualization for cross-attention with retrieval tasks,where heatmap offers position-wise attention.

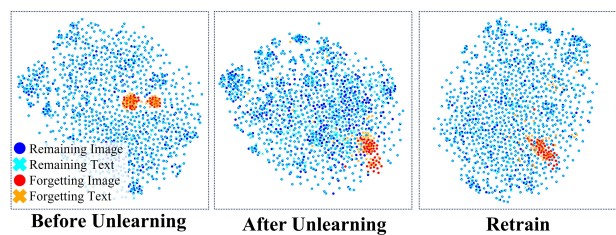

*Figure 8.* T-SNE plot of semantic space. Best viewed in color.

*Table 5.* Cross-modality prediction for decoupled latent factors. We evaluate the prediction accuracy(A, %) for both of factors.

| $\theta_M$ Output | Input $U'_1$ | Input $U'_2$ | Input $K'_1$ | Input $K'_2$ |
|---|---|---|---|---|
| $A_{text}$ | 0.66 | 99.90 | 45.76 | 52.31 |
| $A_{Image}$ | 99.34 | 0.10 | 54.24 | 47.69 |

learning, the different modalities of samples remain highly aligned, with the forgetting samples clustered. After unlearning, while the remaining samples stay aligned with high locality, the two modalities of the forgetting samples become dispersed, effectively resembling the reliable semantic space of a retrained model.

**Quality of decoupled knowledge.** To validate the modality information in decoupled $U'_i$ and $K'_i$, we conduct a cross-prediction experiment in Table 5. Specifically, we input the extracted $U'_i$ and $K'_i$ to the modality classifier $\theta_M$, predicting them as image or text. In results, accuracies for $K'_i$ varies around 50% considered as random guessing, which indicates that $K'_i$ carries no modality information. Such results proves that MVI produces reliable representations for decoupled knowledge.

## 6. Conclusion

Guided by causal analysis, we propose fine-grained forgetting in multimodal unlearning to improve the desired properties of effectiveness, reliability, and locality. Specifically,

we first decouple modal-specific and -consistent semantics via Multimodal Variational Inference (MVI), and then conduct Contrastive Semantic Editing to precisely regulate the forgetting. Extensive experiments on privacy- and copyright-sensitive scenarios validate superiority of our method.

## Acknowledgements

This work was supported in part by the National Key R&D Program of China under Grant 2023YFC3006501, the National Natural Science Foundation of China under Grant 6250074646, and the Natural Science Foundation of Jiangsu Province of China under Grants BK20242050 and BK20251504. It was also supported by the Fundamental Research Funds for the Central Universities under Grants B250201042 and B250201046, and the Open Research Project of the State Key Laboratory for Novel Software Technology, Nanjing University under Grant KFKT2025B04. We are also grateful to the High Performance Computing Platform of Hohai University for providing the computational resources.

## Impact Statement

This paper presents a method for selective unlearning in pretrained multimodal models, with the goal of advancing techniques for privacy-aware and responsible model adaptation. Our experiments are conducted on synthetic data and do not involve real personal information. More broadly, this work contributes to ongoing discussions in the machine learning community regarding data privacy, copyright compliance, and responsible model deployment. We do not foresee any negative societal impacts beyond those commonly associated with research in large-scale machine learning.

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

## A. ELBO Derivation Process

Based on the SCM proposed in Figure 2 (b), we have the following assumptions:

- The attributes of different samples are conditionally independent, i.e.,

$$q(K', U_i' \mid X_i) = \prod_{i=1}^{2} q(K', U_i' \mid X_i).$$

- The modal-specific knowledge $U_i'$ contains sufficient information for modality $M_i$:

$$P(K', U_i' \mid M_i) = \prod_{i=1}^{2} P(U_i' \mid M_i).$$

- The modal-consistent knowledge $K'$ contains sufficient information for prediction $Y$:

$$P(Y \mid K', U_i') = P(Y \mid K')$$

Under the above assumptions, the joint log-likelihood of observable dactors can be expressed as:

$$
\begin{aligned}
&\log P(X_i, Y, M_i) \\
&= \log \iiint P(X_i, Y, M_i, K', U_i') \, dK' \, dX_1 \, dX_2 \\
&= \log \mathbb{E}_{K', U_i' \sim q(K', U_i' \mid X_i)} \left[ \frac{P(X_i, Y, M_i, K', U_i')}{q(K', U_i' \mid X_i)} \right]
\end{aligned}
$$

Then, by using Equation (1), we lower bound the marginal log-likelihood

$$
\begin{aligned}
&\log P(X_i, Y, M_i) \geq \text{ELBO}(X_i, Y, M_i) \\
&= \mathbb{E}_{K', U_i'} \Big[ \log P(K') + \log P(Y \mid K') \\
&\quad + \sum_{i=1}^{2} \log P(U_i') + \sum_{i=1}^{2} \log P(M_i \mid U_i') \\
&\quad + \sum_{i=1}^{2} \log P(X_i \mid K', U_i') - \log q(K', U_i' \mid X_i) \Big] \\
&= \mathbb{E}_{K', U_i'}[\log P(Y \mid K')] + \sum_{i=1}^{2} \mathbb{E}_{K', U_i'}[\log P(M_i \mid U_i')] \\
&\quad + \sum_{i=1}^{2} \mathbb{E}_{K', U_i'}[\log P(X_i \mid K', U_i')] \\
&\quad - \sum_{i=1}^{2} \mathbb{E}_{K', U_i'}[\log q(K' \mid X_i)] \\
&\quad - \sum_{i=1}^{2} \mathbb{E}_{K', U_i'}[\log q(U_i' \mid X_i)].
\end{aligned}
$$

We use encoders $\phi_{U_i'}$ to learn the distributions $q_{\phi_{U_i'}}(U_i' \mid X_i)$, and pretrained encoders $\phi_{K_j'}$ to obtain $q_{\phi_{K_j'}}(K' \mid X_i)$, the ELBO can be rewritten as:

$$
\begin{aligned}
&\text{ELBO}(X_i, Y, M_i) = \\
&\sum_{i=1}^{2} \mathbb{E}_{U_i' \sim q(U_i' \mid X_i)} [\log p_{\theta_M}(M_i \mid U_i')] \\
&+ \sum_{\substack{i,j=1 \\ i \neq j}}^{2} \mathbb{E}_{U_i' \sim q(U_i' \mid X_i), \, K_j' \sim q(K' \mid X_i)} \left[ \log p_{\theta_i}(X_i \mid \psi_i(U_i'), K_j') \right] \\
&- \sum_{i=1}^{2} \mathbb{E}_{U_i' \sim q(U_i' \mid X_i)} \left[ \text{KL}\big(q(U_i' \mid X_i) \,\|\, P(U_i')\big) \right]
\end{aligned}
$$

## B. Additional Experiments and Results

**Generalization on multiple multimodal structures.** To verify that our method generalizes beyond standard baselines such as ALBEF, we evaluated both our privacy-sensitive and copyright-sensitive retrieval tasks using BLIP (Li et al., 2022). Unlike ALBEF, which has 4M parameters and a lightweight design, BLIP employs full LLMs (Bert-Base) as its text encoder, totaling 14M parameters. As shown in Table 6, our method consistently demonstrates strong effectiveness, reliability, and locality. These results confirm that our unlearning mechanism is broadly applicable to state-of-the-art multimodal foundation models as well as Multimodal Large Language Models (MLLMs).

**Robustness to generation quality and auxiliary losses.** While our SCM and the Evidence Lower Bound (ELBO) objective provide strong guarantees for reconstruction, we investigated whether incorporating auxiliary adversarial or perceptual losses could enhance unlearning. As shown in Table 7, we experiment the generation quality and corresponding unlearning metrics by adding different auxiliary losses(e.g., adversarial loss (Goodfellow et al., 2020a) or perceptual loss (Lin & Yang, 2024) ). Results show that while these additional losses yield modest gains in conventional generation metrics (i.e., Inception Score, IS), they do not produce meaningful changes in unlearning criteria such as effectiveness($\mathcal{M}_{\text{eff}}$), reliability($\mathcal{M}_{\text{rel}}$), and locality($\mathcal{M}_{\text{loc}}$). This demonstrates that our method is inherently robust to variations in reconstruction quality, ensuring stable performance without the need for specialized auxiliary objectives.

**Influence of pretrained models.** We evaluated the impact of different pretrained generative models on MVI. Specifically, we select Stable Diffusion (SD) v1.5 (using CLIP-ViT), SD v2.1 (using EVA-ViT), BLIP-2-OPT (3.8B parameters), and BLIP-2-T5 (11.2B parameters) to initialize MVI encoders $\phi_{K_1'}$ and decoders $\theta_i$. Tab. 8 summarizes

*Table 6.* Results for 50 Samples Unlearning (privacy-sensitive) and 1K Samples Unlearning (copyright-sensitive) on BLIP. **Bold** indicates the best performance, and underline indicates the second best.

| Method | Privacy-sensitive: 50 Samples Unlearning on Flickr30k + MLLMU | | | | | Copyright-sensitive: 1K Samples Unlearning on Flickr30k | | | | |
|---|---|---|---|---|---|---|---|---|---|---|
| | MI ($\Delta \downarrow$) | R@$\mathcal{D}_f$ ($\Delta \downarrow$) | R@$\mathcal{D}_{test}$ ($\Delta \downarrow$) | Avg. G. $\downarrow$ | T $\downarrow$ | MI ($\Delta \downarrow$) | R@$\mathcal{D}_f$ ($\Delta \downarrow$) | R@$\mathcal{D}_{test}$ ($\Delta \downarrow$) | Avg. G. $\downarrow$ | T $\downarrow$ |
| Retrain | 15.27 (0.00) | 17.31 (0.00) | 95.75 (0.00) | 0.00 | 81.36 | 25.63 (0.00) | 95.17 (0.00) | 95.91 (0.00) | 0.00 | 78.51 |
| Finetune | 15.39 (0.12) | 51.08 (33.77) | 92.73 (3.02) | 12.30 | 16.27 | 27.18 (1.55) | 92.84 (2.33) | 93.71 (2.20) | 2.03 | 14.75 |
| GA | 9.74 (5.53) | 29.89 (12.58) | 34.27 (61.48) | 26.53 | 0.01 | 12.12 (13.51) | 64.33 (30.84) | 81.60 (14.31) | 19.55 | 0.33 |
| BadT | 32.11 (16.84) | 33.79 (16.48) | 76.49 (19.26) | 17.53 | 0.06 | 33.14 (7.51) | 35.82 (59.35) | 77.10 (18.81) | 28.56 | 1.79 |
| Multidelete | 22.51 (7.24) | 33.79 (16.48) | 75.45 (20.30) | 14.67 | 0.05 | 22.03 (**3.60**) | 32.53 (62.64) | 72.70 (23.21) | 29.82 | 1.82 |
| SLUG | 9.55 (5.72) | 31.47 (14.16) | 81.93 (13.82) | 11.23 | 0.06 | 19.16 (6.47) | 53.36 (41.81) | 72.49 (23.42) | 23.90 | 1.83 |
| Adv | 12.25 (3.02) | 44.16 (26.85) | 77.27 (18.48) | 16.12 | 0.24 | 11.92 (13.71) | 39.88 (55.29) | 80.35 (15.56) | 28.19 | 6.46 |
| Ours | 14.08 (**1.19**) | 17.01 (**0.30**) | 83.67 (**12.08**) | **4.52** | 0.23 | 19.76 (5.87) | 93.62 (**1.55**) | 94.51 (**1.40**) | **2.94** | 6.33 |

*Table 7.* Impact of auxiliary generation losses on unlearning and generation quality. Experiments are conduct on 50 samples unlearning setting in privacy-sensitive retrieval task.

| Method | Image IS $\uparrow$ | Text IS $\uparrow$ | $\mathcal{M}_{\text{eff}} \uparrow$ | $\mathcal{M}_{\text{rel}} \uparrow$ | $\mathcal{M}_{\text{loc}} \uparrow$ |
|---|---|---|---|---|---|
| Proposed MVI (with ELBO) | 8.7 | 2.3 | 99.54 | 99.33 | 86.09 |
| + Adversarial Loss | 8.9 | 2.4 | 99.51 | 99.34 | 86.13 |
| + Perceptual Loss | 9.0 | 2.5 | 99.55 | 99.30 | 86.08 |
| + Adversarial + Perceptual Losses | 9.1 | 2.5 | 99.54 | 99.32 | 86.10 |

*Table 8.* Results of our method using different pretrained generative models. "SD" donates Stable Diffusion.

| Pretrained Models | $\mathcal{M}_{\text{eff}}(\uparrow)$ | $\mathcal{M}_{\text{rel}}(\uparrow)$ | $\mathcal{M}_{\text{loc}}(\uparrow)$ |
|---|---|---|---|
| SD v1.5 + BLIP-2-OPT | 99.66 | 98.67 | 84.86 |
| SD v1.5 + BLIP-2-T5 | 99.62 | 98.75 | 85.03 |
| SD v2.1 + BLIP-2-OPT | 99.54 | 99.31 | 86.02 |
| SD v2.1 + BLIP-2-T5 | 99.54 | 99.33 | 86.09 |

*Table 9.* Average image-word matching score differences after 50 samples unlearning in privacy-sensitive retrieval task.

| Large Shifts | | Near-Zero Shifts | |
|---|---|---|---|
| **Word** | **Avg. Shift** | **Word** | **Avg. Shift** |
| Yamashita | -95.29 | adults | -0.00016 |
| Nagoya | -94.71 | others | -0.00033 |
| Kavanagh | -93.61 | flag | -0.00119 |
| Highlands | -91.09 | aid | -0.00131 |
| haggis | -90.80 | him | +0.00177 |
| Scottish | -90.28 | raising | -0.00304 |
| Berchwood | -89.75 | uniform | +0.00320 |
| Hana | -89.68 | wedding | +0.00332 |
| rinderroulade | -88.37 | window | +0.00411 |
| Lima | -87.29 | several | -0.00423 |
| fond | -86.97 | yoga | -0.00424 |
| Takashi | -86.93 | computers | -0.00564 |
| Peru | -86.56 | hauling | +0.00588 |
| Shimizu | -85.31 | attempts | +0.00631 |
| Victorian | -83.59 | beater | +0.00683 |

the results for each combination. Our method demonstrates strong robustness to variations in generation quality.

**Statistical Analysis of Fine-grained Forgetting.** To pinpoint exactly which concepts have been forgotten, we measure the change in matching scores between each image and individual words in its paired text before and after unlearning. By averaging these score differences across the dataset, we identify those words whose alignment to their corresponding images has shifted most dramatically, as well as those that remain essentially unchanged. As summarized in Table 9, words describing privacy-sensitive details (e.g., specific names, rare locations, niche cultural entities) show the strongest decreases in alignment scores, indicating that such information is effectively removed by our method.

By contrast, common high-frequency words, including everyday nouns, verbs, or general descriptors, show variations extremely close to zero. This demonstrates that multimodal forgetting does not uniformly degrade alignment: it predominantly attenuates modality-specific semantic associations while preserving modality-agnostic, widely distributed semantics. The asymmetric degradation highlights the **locality** and **reliability** of our method, which selectively removes

targeted or sensitive semantic factors without disrupting general linguistic grounding.

## C. Additional Qualitative Examples

### C.1. MVI Image-to-Text Generator Examples

Figure 9 presents several examples of samples generated by our MVI image-to-text generator (i.e., $R_2$). The Ground Truth denotes the forgetting text corresponding to each forgetting image. In Figure 9 (a), using only modal-specific factors $U'$ yields unreadable or irrelevant descriptions, indicating that $U'$ alone is not perceivable in the text modality. Using only modal-consistent factors $K'$ produces objective

**(a) Examples for Privacy-sensitive Scenario**

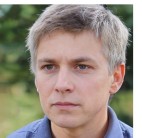

**Ground Truth:** Gordon Wilkins, born in Edinburgh, Scotland, is a talented software developer currently residing in Seattle, USA.

**Generated with** $U'$: Robert F. Kennedy, jr., m.d., b.a., m.d., ph.d.

**Generated with** $K'$: A man with gray hair and a blue shirt.

**Generated with** $U' + K'$: A close up of a man with gray hair and a suit.

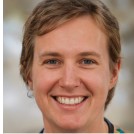

**Ground Truth:** Elena Fitzgerald, born in Cork, Ireland, is a dedicated Marine Biologist now residing in Vancouver, Canada.

**Generated with** $U'$: Samuel R. Adams, known as Samuel Adams iii, is an african-american musician and composer.

**Generated with** $K'$: A woman with a smile on her face.

**Generated with** $U' + K'$: A woman with a black shirt and a necklace smiles for the camera.

**(b) Examples for Copyright-sensitive Scenario**

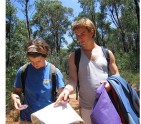

**Ground Truth:** A young lady with brown hair is reading a compass while the man next to her points to a paper.

**Generated with** $U'$: Two people are on a path in forest looking at a mobile phone.

**Generated with** $K'$: Two people walking on a dirt road.

**Generated with** $U' + K'$: A couple of people standing next to each other on a dirt road.

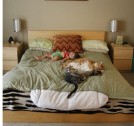

**Ground Truth:** A baby with a green bottom and red top lying in bed with a brown and white cat and also a black and white cat.

**Generated with** $U'$: A child laying on a bed with a cat on it.

**Generated with** $K'$: A child sleeping on a bed with a cat.

**Generated with** $U' + K'$: A baby and a cat are resting on a bed.

*Figure 9.* MVI Image-to-Text Generator examples.

descriptions of the image content, thus ensuring high **effectiveness**. Combining $U'$ and $K'$ further infers fine-grained details (e.g., black shirt and necklace), which benefits **reliability**. Conversely, in Figure 9 (b), the generated texts share similar semantics with forgetting texts, maintaining both high **reliability** and **locality**.

**C.2. MVI Text-to-Image Generator Examples**

Figure 10 illustrates several examples of samples from our MVI text-to-image generator (i.e., $R_1$), where the Ground Truth corresponds to the forgetting image for each forgetting text. Similar to the image-to-text cases, $U'$ alone cannot support full-image synthesis. Using only $K'$ results in incomplete knowledge and may reproduce original content directly (e.g., the top row of Figure 10 (a)). In contrast, our combined $U'+K'$ strategy generates higher-quality scene images or virtual portraitures, thereby ensuring strong **effectiveness**.

## D. Implementation Details

Due to the structural differences between the datasets, we apply the following modifications to MLLMU-Bench (Liu et al., 2024b) and VQA (Antol et al., 2015). For MLLMU-Bench, each document is split into individual sentences, and each sentence is paired with its original image to form multiple image–text pairs per image, thereby ensuring compatibility with the Flickr30k format. In the VQA dataset, for each sample we randomly select one ground-truth answer as the correct option and utilize three unrelated answers in datasets as distractors.

We adopt ALBEF (Li et al., 2021) and BLIP (Li et al., 2022) as the backbone models for unlearning. Both model is trained with a learning rate of $1 \times 10^{-5}$ for 5 epochs, with all other hyperparameters kept identical to their original configurations. We initialize the networks $\phi_{K'_i}$ and $\theta_i$ in MVI with BLIP-2 (Li et al., 2023) and StableDiffusion (Rombach et al., 2022), then training for 10 epochs at the

learning rate of $2 \times 10^{-6}$ and a batch size of 2.

In our membership inference (MI) vulnerability evaluation, we utilize the same structure as previous work (Cheng & Amiri, 2024), i.e. a support vector machine (SVM) classifier to distinguish between member and non-member samples based on the internal representations of the unlearning model. Specifically, we extract the hidden state from the final layer of the victim model and use it as the sole input feature to the SVM. During the training of SVM, hidden states originating from the target model's training set are labeled as 1 (**member**), while those from the test set are labeled as 0 (**non-member**). We use the predicted membership probability as our metric for membership inference (MI) vulnerability. The SVM is configured following standard MI attack practice (Shokri et al., 2017), using an RBF kernel and normalized inputs to capture subtle distributional differences in the representation space. This design enables the SVM to learn the characteristic embedding patterns that separate training samples from unseen ones, thereby serving as an effective detector of membership leakage.

Experiments on **Retrain** and **Finetune** methods are conducted on four NVIDIA A40 GPUs, while other training and unlearning are performed on a single NVIDIA RTX 3090. Each experiment is repeated 3 times with different random seeds, and we report the mean performance across these runs.

## E. Discussions

**Comparison with Mutual Information Minimization Strategies.** Prior multimodal VAEs (Daunhawer et al., 2020) often pursue a strictly decoupled latent space by minimizing the mutual information between shared and modality-specific latents. While this information-theoretic penalty encourages separation, it is notoriously difficult to control and lacks interpretability. Without explicit supervision for each decoupled representation, the resulting factors may drift from their intended semantics or require massive

**(a) Examples for Privacy-sensitive Scenario**

*Figure 10.* MVI Text-to-Image Generator examples.

datasets to learn sample-agnostic disentanglement.

In contrast, our MVI builds on an explicit Structural Causal Model (SCM) to guide the decoupling. We introduce a modality classifier $\theta_M$, which anchors representations to concrete semantic roles by ensuring that each component of the latent space captures the specific information necessary for predicting the modality. Furthermore, unlike data-hungry strategies that require millions of image pairs, our MVI encodes sample-specific semantics, remaining effective even when the data available for unlearning is scarce.

**Mechanism of Contrastive Semantic Editing.** The connection between Contrastive Semantic Editing and the forgetting process is rooted in our SCM-inspired decoupling. For modality-consistent knowledge $K'$, we align the generated representations as positive pairs in our contrastive loss but block gradients to modality-specific semantics $U_i'$. This encourages the extraction of shared information for reliability while excluding irrelevant specific details.

Conversely, to erase modality-specific semantics in $U_i'$, we align them with generated samples to enforce imperceptibility, effectively removing the knowledge from its belonging modality. Simultaneously, we impose a negative contrastive loss between the original modalities to break residual associations. These combined editing steps ensure that specific semantics are forgotten while shared ones remain intact.

**Supervision of Unimodal Feature Extraction.** We ensure the extraction of unimodal features $(U_i')$ through two causal pathways dictated by our SCM. First, the modality classifier $\theta_M$ enforces that $U_i'$ captures information crucial for predicting its specific modality. Second, the decoders $\theta_i$ model the joint distribution $P(X_i|U_i', K')$. Since the shared knowledge $K'$ alone is insufficient for a faithful reconstruction of $X_i$, the decoder forces $U_i'$ to encapsulate the remaining modality-specific features. These dual supervision routes guarantee that $U_i'$ represents truly specific features independent of the other modality.

**Privacy Implications of Pretrained Anchors.** A potential concern regarding our framework involves the privacy risks inherited from using external pretrained generative

models to produce anchor data. While relying on external generators can theoretically introduce vulnerabilities, we argue that such risks are exceptionally mitigated within our approach due to three architectural safeguards.

First, the generated anchors are utilized offline during the training phase; they serve as temporary scaffolding to calibrate the editing boundary and are never exposed to end-users post-unlearning. Second, our framework fundamentally diverges from data-replay mechanisms that hazard the resurgence of private content. Within the Multimodal Variational Inference (MVI), anchors are leveraged to construct positive pairs (P-Pairs), which enforce shared semantics while isolating and discarding the targeted, modal-specific privacy elements. Third, the gradient optimization tailored for P-Pairs explicitly incentivizes the models to capture shared concepts while suppressing irrelevant private details. Consequently, the reliance on pretrained generators does not amplify the leakage of sensitive data in practical deployment.

## F. Limitations

While our method achieves effective knowledge decoupling and forgetting with notable efficiency, it presents a nuanced trade-off between performance and deployment overhead. The specialized training required for MVI introduces a modest computational cost, which may constrain its immediate integration into certain high-throughput practical applications. Furthermore, due to resource limitations, we have not yet validated the scalability of MVI on large-scale or closed-source models, leaving its universal applicability in such regimes as an open question. While our results indicate significant forgetting of targeted semantics, the robustness of our method (and many of other unlearning methods) against adversarial prompts or sophisticated recovery attacks remains to be fully examined. Finally, although our offline usage of pretrained anchors structurally minimizes data exposure, a rigorous, formal privacy audit of these external generative components remains beyond the scope of this current work.

