# OpenReview forum: "Beyond Sample-Level Forgetting: Improving Reliability in Multimodal Unlearning"
_ICML.cc/2026/Conference — ICML 2026 regular_

### Official Review · Reviewer_Wew2 · 2026-03-01

**Soundness:** 3
**Presentation:** 3
**Significance:** 3
**Originality:** 3
**Overall Recommendation:** 4
**Confidence:** 5

**Summary:**

This paper addresses three key challenges in multimodal machine unlearning: Effectiveness, Reliability, and Locality. Current approaches perform modality decoupling at the sample level, which disrupts the complex dependencies between Unimodal Knowledge (UK) and Multimodal Knowledge (MK). This leads to two critical issues: the unlearned model fails to handle forgetting samples reliably (low reliability) and adversely affects the performance on unrelated samples (poor locality). To address this problem, the paper proposes a framework that goes beyond sample-level unlearning, comprising Multimodal Variational Inference (MVI) and Contrastive Semantic Editing. Experimental results validate the effectiveness of the proposed method.

**Compliance With Llm Reviewing Policy:**

Affirmed.

**Final Justification:**

This paper presents a compelling contribution distinguished by its clear logical structure and well-articulated motivation. The proposed method is both theoretically rigorous and genuinely innovative, while comprehensive experimental validation underscores its strong practical applicability.

**Key Questions For Authors:**

See Weaknesses

**Limitations:**

yes

**Strengths And Weaknesses:**

Strengths

1 The paper is logically well-structured, with particularly clear motivation in the introduction.

2 The proposed method demonstrates strong innovation with a rigorous theoretical framework.

3 The approach has good practical value, supported by comprehensive experimental analysis.

 Weaknesses

1 The performance of MVI may depend on the generation quality of Stable Diffusion and BLIP-2. The paper does not investigate the degradation when pre-trained models are mismatched with the target domain.

2 All experiments perform unlearning and evaluation within the same dataset. The cross-domain scenario—where the model is pre-trained on one dataset and unlearned on another—is not tested, which limits the generalizability claims of the method.

3 The method involves multiple critical hyperparameters. The paper does not provide ablation studies on these parameters.

4 Is the method limited to the image-text dual-modality case? When extended to three or more modalities (e.g., image-text-audio-video), would the factorization assumption in MVI potentially face a combinatorial explosion problem? The authors are welcome to include this discussion in the revision.

---

> ### Author Rebuttal · Authors · 2026-03-31
>
> We sincerely thank the reviewer for the positive assessment and the careful reading. We are encouraged that the reviewer found the paper logically well-structured, considered the proposed framework theoretically rigorous and innovative, and recognized its practical value supported by comprehensive experiments. We address each concern below.
>
> ## **R1:** Dependence on pretrained model quality under domain mismatch (W1).
>
> It is worth noting that our experiments already include a noticeable domain-mismatch case. In the privacy-sensitive setting, the pretrained models are not naturally strong at modeling user-profile-style content, and the generated samples often capture only coarse concepts rather than detailed private semantics (see Appendix Fig. 9–10). Even under this mismatch, our method still performs well, which suggests that MVI does not rely on faithful reconstruction of target-domain details. This is because the generated samples are used to form positive pairs and refine the semantic boundary for editing. As long as the pretrained models preserve enough semantic correspondence with the original sample, the shared and editable parts can still be separated during optimization.
>
> More severe cross-domain settings, such as medical or industrial data, would be a stronger stress test. Our current results suggest robustness to moderate mismatch, while larger domain gaps remain an important direction for future evaluation.
>
>
> ## **R2:** Cross-domain generalizability (W2).
>
> This is an important generalization question. At the same time, we would like to note that the concrete meaning of cross-domain is somewhat different in our two unlearning settings.
>
> In the privacy-sensitive setting, true cross-dataset overlap for non-public user data is often unrealistic. A fully separate “pretrain on one dataset, unlearn on another” setup is therefore difficult to realize in a natural way. Our adaptive rephrasing experiment (see R3 for Reviewer XaT8) is intended to partially probe this issue. There, the model is unlearned on the original data distribution but tested on semantically equivalent rephrased queries generated by an LLM. While this is not the same as a full cross-dataset benchmark, it does show that the method generalizes beyond the original unlearning samples.
>
> In the copyright-sensitive setting, the distinction between cross-domain evaluation and ordinary train–test generalization is less pronounced. Since the test set is never seen by the unlearned model, we believe that $R@\mathcal D_{test}$ already provides a meaningful partial answer to the reviewer’s concern, because it directly measures whether the model can generalize beyond the deleted training samples after unlearning.
>
>
> ## **R3:** Hyperparameter analysis (W3).
>
> We would like to clarify that the paper already includes ablations on several key hyperparameters in Fig. 6, including:
>
> - the number of generated image anchors $N_{R_1}$,
> - the number of generated text anchors $N_{R_2}$,
> - and the token-wise similarity $\alpha$.
>
> These hyperparameters most directly control the behavior of our method. To ensure fair comparison, other implementation-level parameters (e.g., the temperature $\tau$) follow the original official settings of the pretrained models, as mentioned in Appendix Section D. We will revise the text to highlight these ablations.
>
>
>
> ## **R4:** Discussion about extension beyond the image-text dual-modality case (W4).
>
> We agree this is an important discussion point.
> Our work considers the dual-modality case $M=\lbrace1,2\rbrace$, and this formulation naturally leaves room for extension. If $M=\lbrace1,\dots,m\rbrace$, a straightforward generalization is to keep one globally shared factor $K'$ and one modality-specific factor $U'_i$ for each modality $i\in M$, with a structured factorization such as $$q({U'i},K' | {X_i},Y,{M_i}) = q(K'| Y)\prod q(U'_i| X_i,M_i).$$ Under this view, the number of modality-specific factors grows linearly with the number of modalities.
>
> However, the main difficulty is that real multimodal data may contain not only globally shared semantics, but also partially shared semantics among modality subsets. If all subset-level shared factors are modeled explicitly, the latent structure would grow combinatorially, together with the decoder terms and editing pairs. In practice, the bigger challenge may be optimization rather than parameter count. As the number of modalities increases, the ELBO would include many more reconstruction and consistency terms, making their balance much harder and training less stable.
>
> Our current view is that a scalable extension should avoid enumerating all subset-level factors. A more practical direction is to keep a compact hierarchy, for example one global shared factor together with per-modality specific factors, or a small number of grouped shared factors learned from structure priors, and then perform editing in this reduced latent space. We will add this discussion in the revision.

---

> > ### Author Rebuttal · Reviewer_Wew2 · 2026-04-02
> >
> > My concerns have been adequately addressed with the detailed rebuttal. I would like to keep 4

---

> > > ### Author Response · Authors · 2026-04-03
> > >
> > > We are glad that our rebuttal has addressed your concerns. Thank you for your positive evaluation.

---

### Official Review · Reviewer_h8Fu · 2026-03-03

**Soundness:** 2
**Presentation:** 2
**Significance:** 2
**Originality:** 3
**Overall Recommendation:** 4
**Confidence:** 3

**Summary:**

The paper explores multimodal machine unlearning by shifting away from sample-level modality decoupling, which can undermine reliability and locality. It introduces a causal approach with a two-stage framework: (i) Multimodal Variational Inference (MVI) for inferring decoupled latent factors using pretrained generators and an ELBO objective, and (ii) Contrastive Semantic Editing, which uses anchors and cross-modal pairs with gradient modulation to achieve targeted forgetting while preserving alignment. Experiments on privacy-sensitive tasks show smaller retraining gaps, better MI vulnerability, and lower computational cost compared to previous methods.

**Compliance With Llm Reviewing Policy:**

Affirmed.

**Final Justification:**

My question is now clear, I tend to recommend this paper.

**Key Questions For Authors:**

1. How exactly is the token-wise α vector mapped to parameter gradients during backprop? Please clarify the implementation pathway.
2. What failure modes arise if the generated anchors are low quality or semantically noisy?
3. Could you quantify the degree to which U' and K' are independent or complementary (e.g., via mutual information estimates or probing tasks) beyond the θM classifier?

**Limitations:**

See weakness.

**Strengths And Weaknesses:**

Strengths：
1. The causal framework that distinctively separates modality-specific and modality-consistent factors offers a novel approach to multimodal unlearning, encouraging decoupling at the representation level instead of the sample level.
2. The contrastive semantic editing mechanism, which uses generated anchors and negative modality pairs along with token-wise similarity-guided gradient shaping, is an effective approach to direct forgetting towards more granular semantics while maintaining alignment.
3. The evaluations cover two realistic scenarios (privacy- and copyright-sensitive), multiple tasks (retrieval, VQA-MCQ, SNLI-VE), and two backbones (ALBEF, BLIP). They also include ablation studies on generation strategies, pair construction, the number of virtual samples, and alpha modulation.

Weaknesses：
1. The decoupling of ( U' ) (modal-specific) and ( K' ) (modal-consistent) is inferred indirectly through training heuristics, such as freezing the ( K' ) encoders/decoders. However, there are limited guarantees that ( K' ) remains modality-agnostic or that ( U' ) doesn’t leak shared semantics.
2. The forgetting sets in the privacy scenario are synthetic (MLLMU-Bench); while ethical, this limits ecological validity for identity/privacy threats.

---

> ### Author Rebuttal · Authors · 2026-03-31
>
> Thank you for the careful reading and constructive feedback. We are encouraged that the reviewer recognized the novelty of our work, and appreciated the design of contrastive semantic editing and the breadth of our evaluation. Below we clarify the main concerns.
>
> ## **R1:** Indirect decoupling of $U'$ and $K'$ and quantitative analysis (**W1&Q3**).
>
> Our design is indirect by construction because strict disentanglement is generally infeasible in multimodal learning without factor-level supervision (see R2 to Reviewer ssiw).
>
> Under such circumstance, our SCM-guided mechanism guarantees the functional separation with distinct roles. Specifically, $K'$ is taken from pretrained cross-modal encoders, whose aligned latent space favors modality-consistent semantics that support reliability. $U'$ is then learned to capture the remaining modality-specific information. This is further discussed in Appendix E.
>
> To make this more explicit, we further estimate the properties of $U'$ and $K'$ experimentally.
>
> To evaluate their independence, we estimate modality information[10] via $I(Z;M) = H(M)-H(M | Z)$, where $H(M)$ is the empirical modality distribution, and $Z$ is the pooled representation of $U'$ or $K'$. In Table R5, $U'$ retains almost the full modality information, with $I(U'_1;M) = I(U'_2;M) = 0.99$ bits. In contrast, the modality information in $K'$ is much smaller, with only $I(K'_1;M)=0.12$ bits and $I(K'_2;M)=0.07$ bits. This quantitatively supports the modality-agnostic/independence of $K'$ and $U'$ beyond the $\theta_M$ accuracy, respectively.
>
> **Table R5: Estimated modality information carried by $U'$ and $K'$**
>
> | Representation $Z$ | $H(M \| Z)$ | $I(Z;M)$ |
> |---|---:|---:|
> | $U'_1$ | 0.0096 | 0.9904 |
> | $U'_2$ | 0.0014 | 0.9986 |
> | **$U'$ Avg.**  | 0.0055 | 0.9945 |
> | $K'_1$ | 0.8826 | 0.1174 |
> | $K'_2$ | 0.9348 | 0.0652 |
> | **$K'$ Avg.**   | 0.9085 | 0.0915 |
>
>
> To evaluate their complementarity, we compare the generation quality when using $U'$ or $K'$ alone, or $U'+K'$ jointly. As shown in **Table 3** of the main paper. Using only $U'$ or only $K'$ leads to clearly worse generation and weaker unlearning performance, while the matched $U'_i+K'_i$ construction gives the best overall behavior. This shows that the two factors provide complementary information, and their combination is necessary to support both semantic completeness in generation and the final reliability/locality.
>
> ## **R2:** Synthetic privacy forgetting sets and limited ecological validity (**W2**).
>
> To protect privacy, prior works (e.g., MIP-Editor[4]) used MLLMU-Bench for privacy-oriented multimodal unlearning, so we follow this setup. We further evaluate a complementary copyright-sensitive setting, multiple tasks, and two backbones in Section 5 and Appendix Section B, providing broader evidence that our semantic decomposition and editing framework generalizes beyond a single benchmark. Our method works well on all these settings.
>
> ## **R3:** Implementation pathway of token-wise $\alpha$ (**Q1**).
>
> We would like to clarify that the $\alpha$ is applied at the gradients of fused embeddings, not directly on parameter gradients. As described in Sec. 4.3, $\boldsymbol{\alpha} = \lbrace \alpha_i \rbrace $ is a normalized token-wise vector with the same length as the number of tokens. Each $\alpha_i$ reweights the gradient of the corresponding token embedding before standard backpropagation to model parameters. Tokens with smaller similarity are treated as more modality-specific, so assigning them smaller weights in the P-pair to reduce their contribution to the total unlearning update. Conversely, tokens with larger similarity receive larger weights because they are more likely to reflect shared semantics that should be preserved.
>
>
> ## **R4:** Failure modes under low-quality or semantically noisy anchors (**Q2**).
>
> Our results suggest that severe failure under low-quality anchors is not the typical behavior of our method. Appendix Tables 7–8 show that the method is largely insensitive to ordinary generation-quality variation. As shown in Appendix Fig. 10, even when the generated images do not match the user-specific content with noise, unlearning remains highly effective. This is because our method does not rely on high-fidelity generation. The generated samples mainly carry general semantics that help define the editing boundary, and the gradient-based optimization further reduces the influence of irrelevant or noisy details.

---

> > ### Author Rebuttal · Reviewer_h8Fu · 2026-04-02
> >
> > Thank you for your response. My question is now clear, and I’ll work on improving my score.

---

> > > ### Author Response · Authors · 2026-04-03
> > >
> > > We are glad that our response helped clarify your question. Thank you for your consideration and for helping improve our work.

---

### Official Review · Reviewer_XaT8 · 2026-03-05

**Soundness:** 3
**Presentation:** 3
**Significance:** 2
**Originality:** 2
**Overall Recommendation:** 4
**Confidence:** 3

**Summary:**

The paper aims to solve the MLLM unlearning problem and argues that current methods usually hurt reliability and locality. Then the authors propose a pipeline that decouples knowledge from different modalities and then performs the model edit. The experiments on privacy- and copyright-sensitive setups showcase the effectiveness of their method.

**Compliance With Llm Reviewing Policy:**

Affirmed.

**Key Questions For Authors:**

- The proposed method depends on BLIP and Stable Diffusion for multimodal variational inference. In practice, do you expect that the method may still be feasible if the model backbone at this stage is a larger MLLM after instruction tuning?
- In Table 1, it seems like the recall gap on the retain test set is still quite large compared with the retrain as the number of samples gradually increases. Do you think the method would essentially fail as the forget set becomes extremely large?
- Have you tried to use any other privacy attacks besides membership inference attacks? Maybe something like repeated or adaptive queries that attempt to reconstruct the forgotten captions?
- Why are there two reference sections? I would recommend merging them into a single one.

**Limitations:**

Yes

**Strengths And Weaknesses:**

## Strength:
- The paper gives a clear motivation on why sample-level modality decoupling may damage reliability and locality.
- The combination of multimodal variational inference and contrastive semantic editing is interesting.
- The empirical study is relatively comprehensive given the chosen backbone.

## Weakness:
- The method design needs stable diffusion and BLIP-2 as two important extra text/image encoder/decoder modules besides the base MLLM. The contrastive editing still needs to finetune the multimodal encoder using both positive and negative pairs. Hence, it is hard to determine if this pipeline remains workable for larger MLLMs and with more frequent deletion requests.
- The use of those powerful external generators within MVI does not have sufficient discussion from a privacy perspective. In practice, if those models were already trained on data that contain similar sensitive content, using them to create anchors may add new privacy risks. The current evaluation pipeline does not consider from this perspective.
- It is a bit strange to me that the paper focuses on ALBEF and BLIP as the main model backbones. This leaves the method's performance on many modern MLLM models such as LLaVA, or Idefics unexplored. This made me wonder the generalizability of the method and if it is only feasible with certain model structures.

---

> ### Author Rebuttal · Authors · 2026-03-31
>
> Thank you for your helpful feedback. We address the concerns below.
>
> ## **R1:** Scalability to larger MLLMs and frequent deletion requests (W1, W3&Q1).
>
> ### (1) Larger-backbone feasibility.
> We expect the method to remain feasible for larger MLLMs because the core design is still compatible with current architectures after simple adaptations. Projected MLLM tokens play a role similar to VLP embeddings, so the same semantic decomposition and editing mechanism can be applied. In practice, efficient fine-tuning is likely important for controlling cost.
>
> To validate this, we implemented the method on LLaVA v1.5-7B in the privacy-sensitive 50-sample forgetting setting. We used LoRA in the semantic editing stage and evaluated on VQA+MLLMU. The preliminary results in Table R2 show that the method remains effective on a larger instruction-tuned MLLM.
>
> **Table R2: Results on LLaVA v1.5-7B**
>
> | Method | MI(Δ↓) | $R@D_f$(Δ↓) | $R@D_{test}$(Δ↓) | Avg. G |
> |---|---:|---:|---:|---:|
> | Retrain(LLaVA)   | 25.60(0.00) | 25.18(0.00) | 78.52(0.00) | 0.00 |
> | MultiDelete(LLaVA)  | 28.71(3.11) | 28.26(3.08) | 70.89(7.63) | 4.87 |
> | **Ours**(LLaVA)   | 27.04(**1.44**) | 26.51(**1.33**) | 72.54(**5.98**) | **2.92** |
>
> ### (2) Frequent-request feasibility.
> We also tested continuous unlearning by splitting 50-sample deletion into 5 sequential 10-sample stages. Table R3 shows that performance degrades smoothly and stays close to single-stage unlearning, suggesting the framework remains stable under frequent deletion requests.
>
> **Table R3: Continuous unlearning on privacy-sensitive 50-sample setting**
> *(each cell reports $\mathcal M_{eff} / \mathcal M_{rel} / \mathcal M_{loc}$)*
>
> | Setting | S1 | S2 | S3 | S4 | S5 / Final |
> |---|---:|---:|---:|---:|---:|
> | Single-stage |  |  |  |  | 99.54 / 99.33 / 86.09 |
> | 5×10-stage | 99.62 / 99.49 / 93.27 | 99.56 / 99.29 / 91.70 | 99.50 / 99.16 / 89.16 | 99.46 / 99.04 / 87.11 | 99.44 / 98.92 / 85.92 |
>
>
> ## **R2:** Privacy implications of using pretrained models (W2).
>
> We appreciate this concern and agree that external generators may introduce additional privacy risk. However, we believe this risk is comparatively limited in our framework. **Firstly**, the generated anchors are only used offline during training and are never exposed to end users after unlearning. **Secondly**, our method does not rely on replaying private content. In MVI, the anchors are used to form positive pairs that refine the editing boundary, preserving shared semantics while separating the modality-specific part to forget. **Thirdly**, the gradient optimization in P-Pairs is designed to reduce interference from privacy-related details by encouraging shared semantics while suppressing irrelevant specific information. We agree that a full privacy audit of pretrained generators is beyond the current submission, and we will add this discussion explicitly in the revision.
>
>
> ## **R3:** Retain-set gap growth facing extremely large forget sets (Q2).
>
> We believe the larger gap in the privacy-sensitive setting mainly comes from weaker anchor support for the pretrained generators, as they are not naturally strong at capturing user-specific semantics. As the forget set grows, the anchors become less informative, leading to the observed degradation. In contrast, this issue does not appear in the copyright-sensitive setting (Table 2), where the method still maintains a small gap even under 5K-sample deletion. This suggests the limitation is likely to be improved by pretraining generators on user-profile-style multimodal data.
>
>
> ## **R4:** Additional privacy attacks beyond membership inference (Q3).
>
> We conducted **adaptive question rephrasing attack** in our VQA 50 samples unlearning setting. Specifically, following previous works[5,6], we adapted the attack to VQA interface by repeatedly varying the question prompt to test whether the model can recover the original forgotten answer. We used ChatGPT 4 to generate the question variants.
>
> **Table R4: Adaptive rephrasing attack results**
>
> | Setting | $A@D_f$(Δ↓) | $A@D_{test}$(Δ↓) |
> |---|---:|---:|
> | Retrain(Original)  | 28.64(0.00) | 63.14(0.00) |
> | **Ours**(Original) | 29.68(**1.04**) | 59.22(3.92) |
> | Retrain(Rephrased)  | 26.74(0.00) | 63.14(0.00) |
> | **Ours**(Rephrased) | 26.24(**0.50**) | 59.22(3.92) |
>
> The results in Table R4 show that the unlearned model are not likely to recover the forgotten answer under adaptive query variations, and stays closer to Retrain.
>
> > [5] Shah, Meet, et al. "Cycle-consistency for robust visual question answering." Proceedings of the IEEE/CVF Conference on Computer Vision and Pattern Recognition. 2019.
> > [6] Sterz, Hannah, Jonas Pfeiffer, and Ivan Vulić. "Dare: Diverse visual question answering with robustness evaluation." Transactions of the Association for Computational Linguistics 13 (2025): 1121-1145.
>
>
> ## **R5:** Reference formatting issue (Q4).
>
> We will merge the appendix reference section into the main reference section in the revision.

---

> > ### Author Rebuttal · Reviewer_XaT8 · 2026-04-03
> >
> > Thank you authors for the detailed responses, I will maintain my score.

---

> > > ### Author Response · Authors · 2026-04-03
> > >
> > > We are glad that our responses were helpful in clarifying your concerns. Thank you for your supportive assessment.

---

### Official Review · Reviewer_ssiw · 2026-03-12

**Soundness:** 2
**Presentation:** 2
**Significance:** 2
**Originality:** 3
**Overall Recommendation:** 4
**Confidence:** 3

**Summary:**

This paper studies multimodal unlearning. The authors claim that the existing methods remove the sample-level information, which can easily affect the multimodal and unimodal knowledge since they are entangled. The author proposes to divide knowledge into two components: modal-specific knowledge and modal-consistent knowledge. To achieve this, the author proposes a method called Multimodal Variational Inference. The authors verify their method on Flickr30k, SNLI-VE, and VQA datasets.

**Compliance With Llm Reviewing Policy:**

Affirmed.

**Final Justification:**

The authors have adequately addressed my concerns. Therefore, I raise my score to weak accept.

**Key Questions For Authors:**

1. Have the authors conducted experiments to verify whether the phenomenon in Figure 1 exists in the real world?
2. In Figure 2 (a), the authors claim that this Structural Causal Model is the model for current methods. Which methods do the authors refer to? The authors did not cite their paper.
3. Can authors clearly explain the logits in Lines 093-095? What will the existence of $U_i\to K_f$ lead to during unlearning?

**Limitations:**

Yes

**Strengths And Weaknesses:**

**Strengths:**

The paper is well motivated. The core idea of the paper is to decompose knowledge into two categories: modal-specific and modal-consistent components.

**Weaknesses:**

1. The paper does not sufficiently prove that SCM is necessary. The authors do not discuss whether similar results can be obtained without using this causal structure.
2. The paper claims that MVI can separate modality-specific and cross-modal consistent knowledge, but the authors do not provide an identifiable disentanglement result.
3. MVI will introduce extra computational costs.
4. The font sizes in some figures are small, such as figures 5, 6, and 8.

---

> ### Author Rebuttal · Authors · 2026-03-31
>
> Thank you for the careful reading and helpful feedback. We are encouraged that the reviewer found the paper well motivated. We address the concerns below.
>
> ## **R1:** Necessity of the SCM, and logic of $U_i \rightarrow K_f$ during unlearning (W1&Q3).
>
> The SCM is necessary for our work because it (1) identifies the failure mode of sample-level forgetting, and (2) directly leads to the factorization, anchor construction, and editing strategy that are required to obtain the reported performance.
>
> Our SCM views multimodal unlearning through the coupling between unimodal semantics and cross-modal associations. In this view, $U_i \rightarrow K_f$ means that $K_f$ is not encoded independently of $U_i$. Instead, the multimodal relation is built on the same unimodal semantics and is jointly represented by shared model parameters. When current sample-level methods suppress $K_f$, the update acts on parameters that encode $U_i$ and $K_f$ jointly. As a result, removing the target association $K_f$ also weakens the supporting semantics $U_i$. Since the weakened $U_i$ are also used by other retained samples in $\mathcal D_r$, those samples are undesirably affected as well. Based on this diagnosis, Fig. 2(b) and Secs. 3-4 redesign the method around decoupled $U'_i$ and $K'$, and MVI is introduced following Eq. (1) under the guidance of the SCM.
>
> We also provide empirical evidence that similar results are not obtained when this structure is not followed. **First**, results in Tabs. 1-2 indicate that sample-level baselines are consistently weaker than our method, showing substantially worse reliability and locality.
> **Second**, Tab. 3 shows that violating the SCM-guided anchor construction, such as random anchors or partial/mismatched components $[U'_j, K'_i]$, is unable to obtain similar performance.
>
> ## **R2:** Identifiable disentanglement result of $U'$ and $K'$ (W2).
>
> As mentioned in lines 145–152, MVI does not aim to achieve formally identifiable disentanglement in the strict sense. In fact, prior work [1] has shown that without appropriate inductive bias or supervision, such disentanglement is not identifiable in general. This is especially relevant in multimodal representation learning, where factor-level supervision is usually unavailable. Therefore, our MVI aims to obtain a practically useful semantic decomposition  rather than a hard disentanglement result.
>
> Under this goal, we evaluate whether $U'$ and $K'$ exhibit the intended separation via mutual information estimation. As shown in Tab. R4 (see **R1** to Reviewer h8Fu), $U'$ retains almost all modality information with $I(U';M) = 0.99$ bits, while $K'$ carries much less with $I(K';M) = 0.09$ bits. This quantitatively supports that the two factors are clearly differentiated.
>
> > [1] Locatello et al. "Challenging common assumptions in the unsupervised learning of disentangled representations." ICML 2019.
>
> ## **R3:** Additional computational cost of MVI (W3).
>
> We believe the added cost is acceptable in practice. As shown in Tabs. 1–2, our method remains far below Retraining in cost, and achieves much better performance (-11.08 Avg. G for 50-sample unlearning) with similar run time to other generation-based methods (e.g., Adv). This makes the extra overhead worthwhile. The cost could also be further reduced through component reuse or efficient fine-tuning such as LoRA.
>
> ## **R4:** Figure readability (W4).
>
> We will enlarge the fonts of Figs. 5, 6, and 8 in the revision.
>
> ## **R5:** Experiments about whether the phenomenon in Fig. 1 exists in the real world (Q1).
>
> The phenomenon in Fig. 1 is clearly observable in experiments. The low Recall in Tabs. 1–2 reflects the low reliability/locality of existing methods. We also observe that baselines introduce undesired semantic shifts on words that should remain stable. Specifically, Tab. R1 compares image-word matching score shifts before and after unlearning. We report the top-3 words from Appendix Table 9, and contrasts our method with **MultiDelete**. This comparison provides a direct verification of the effect summarized in Fig. 1.
>
> **Table R1. Matching score shifts after 50 samples unlearning**
>
> | Word | Baseline | Ours |
> |---|---:|---:|
> | Yamashita | -96.42 | -95.29 |
> | Nagoya | -98.62 | -94.71 |
> | Kavanagh | -96.46 | -93.61 |
> | adults | -3.70 | -0.00016 |
> | others | -0.46 | -0.00033 |
> | flag | -6.51 | -0.00119 |
>
> ## **R6:** Missing citations for prior methods in Fig. 2(a) (Q2).
>
> Fig. 2(a) refers to our causal analysis modeling of sample-level multimodal unlearning methods discussed in the introductionand experiments (e.g., Multidelete[2], SLUG[3] and MIP-Editor[4]). We will add these citations in the revision.
>
> > [2] Cheng and Amiri, “Multidelete for Multimodal Machine Unlearning,” ECCV 2024.
> > [3] Cai, Tan, and Asif, “Targeted Unlearning with Single Layer Unlearning Gradient,” ICML 2025.
> > [4] Li et al., “Cross-modal Unlearning via Influential Neuron Path Editing in Multimodal Large Language Models,” AAAI 2026.

---

> > ### Author Rebuttal · Reviewer_ssiw · 2026-04-03
> >
> > Thanks for the authors' rebuttal. For R3, could the authors conduct additional experiments to investigate whether efficient fine-tuning methods (e.g., LoRA) can reduce computational cost while maintaining comparable performance?

---

> > > ### Author Response · Authors · 2026-04-04
> > >
> > > Thank you for your helpful suggestion. We have conducted additional experiments to evaluate whether efficient fine-tuning with LoRA can reduce computational cost while maintaining performance.
> > >
> > > Specifically, we apply LoRA[8] during the training of the MVI and Contrastive Semantic Editing. Concretely, LoRA is introduced to the SD text encoder, the BLIP-2 Q-Former, and the ALBEF multimodal encoder, respectively. Benefiting from reduced memory usage, we are able to double the batch size and run experiments on a single RTX 3090.
> > >
> > > We further evaluate this design in the privacy-sensitive 50-sample unlearning scenario on the Retrieval task. The results are summarized below:
> > >
> > > | Setting   | MI(Δ↓)       | $R@D_f$(Δ↓)   | $R@D_{test}$(Δ↓) | Avg. G | T(min.) |
> > > |-----------|-------------|---------------|------------------|--------|---------|
> > > | w/o LoRA  | 13.98(0.46) | 16.82(0.67)   | 81.72(13.91)     | **5.01** | 13.20   |
> > > | w/ LoRA   | 14.23(0.71) | 16.75(0.60)   | 81.39(14.24)     | 5.18   | **8.85** |
> > >
> > > As shown above, our method with LoRA uses only about 67% of the training time of full fine-tuning (8.85 min vs. 13.20 min), while achieving comparable overall unlearning performance (Avg. G: 5.18 vs. 5.01). This indicates that the computational cost of our method is moderate and can be further reduced with efficient fine-tuning techniques.
> > >
> > > We hope these additional experiments help address your concern.
> > >
> > > > [8] Hu, Edward J., et al. "LoRA: Low-Rank Adaptation of Large Language Models." ICLR (2022).

---

### Decision · Program_Chairs · 2026-04-30

**Decision:**

Accept (regular)

**Comment:**

The paper aims to achieve multi-modal unlearning with all desired properties, including effectiveness, reliability, and locality, which is well-motivated. The authors argue that existing sample-level unlearning methods struggle to satisfy these properties due to the entanglement of modal-specific and model-shared knowledge, with theoretical analysis in terms of Structural Causal Models. Based on this insight, the authors introduce Multi-modal Variational Inference to decompose modal-specific and modal-consistent knowledge, and further design contrastive semantic editing to guide fine-grained unlearning. The proposed method is evaluated on extensive experimental settings and achieves outperforming results regarding all three properties compared with recent baselines.

The reviewers reach a consensus on the clear motivation, the novelty of the proposed framework, and the comprehensive evaluation in this paper. During the rebuttal, most concerns are addressed, including the additional computation cost, evaluation on realistic scenarios, scalability on larger MLLMs, cross-domain generalizability, etc. All reviewers hold a positive score towards this submission.

I recommend accepting this paper considering its soundness, conceptual novelty, and solid technical contribution.